# Self-explaining deep models with logic rule reasoning

**Seungeon Lee**[*]
KAIST School of Computing
IBS Data Science Group
archon159@kaist.ac.kr

**Xiting Wang**[†]
Social Computing Group
Microsoft Research Asia
xitwan@microsoft.com

**Sungwon Han**[*]
KAIST School of Computing
IBS Data Science Group
lion4151@kaist.ac.kr

**Xiaoyuan Yi**
Social Computing Group
Microsoft Research Asia
xiaoyuanyi@microsoft.com

**Xing Xie**
Social Computing Group
Microsoft Research Asia
xing.xie@microsoft.com

**Meeyoung Cha**[†]
IBS Data Science Group
KAIST School of Computing
mcha@ibs.re.kr

## Abstract

We present SELOR, a framework for integrating self-explaining capabilities into a given deep model to achieve both high prediction performance and human precision. By "human precision", we refer to the degree to which humans agree with the reasons models provide for their predictions. Human precision affects user trust and allows users to collaborate closely with the model. We demonstrate that logic rule explanations naturally satisfy human precision with the expressive power required for good predictive performance. We then illustrate how to enable a deep model to predict and explain with logic rules. Our method does not require predefined logic rule sets or human annotations and can be learned efficiently and easily with widely-used deep learning modules in a differentiable way. Extensive experiments show that our method gives explanations closer to human decision logic than other methods while maintaining the performance of deep learning models.

## 1 Introduction

Deep learning has shown high predictive accuracy in a wide range of tasks, but its inner working mechanisms are obscured by complex model designs. This raises important questions about whether a deep model is ethical, trustworthy, or capable of performing as intended under various conditions [1].

Many approaches have been proposed to help humans assess and comprehend model decisions. Recent work on explainability has primarily focused on providing **post-hoc** explanations for black-box models that have already been trained [2, 3, 4, 5, 6, 7, 8, 9, 10]. Post-hoc methods do not change the model and hence preserve the predictive performance while providing the additional benefit of explainability. These methods have achieved considerable success in providing valuable insights for model understanding, but there are also known challenges such as computational cost [11] and trust issues [12]. For example, many popular post-hoc methods test the complex black-box model thousands of times to obtain a complete and faithful understanding of the model around a single instance [1, 13, 14]. Subroutines such as full optimization or reverse propagation are generally required, introducing approximations or heuristic assumptions that may lead to misinterpretation [14, 15]. Because there is no guarantee that explanations are always faithful to the model [12], there exists a "general uneasiness" among practitioners about using and trusting post-hoc explanations [16]. **Self-explaining** models naturally solve these issues, making them an ideal choice when interpretability

---

[*]Work done during internship at Microsoft Research Asia
[†]Corresponding Author

36th Conference on Neural Information Processing Systems (NeurIPS 2022).

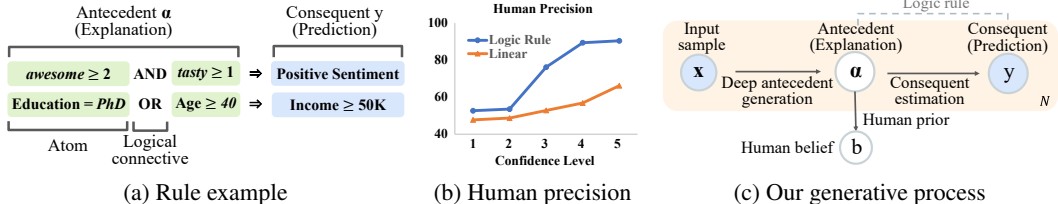

(a) Rule example        (b) Human precision        (c) Our generative process

Figure 1: Reasoning with logic rules: (a) examples of logic rule explanations; (b) human precision for logic rule and linear regression explanations [11]; (c) generative process of our logic rule reasoning.

can be considered from the model design phase [17, 18, 19, 20]. Because the explanation mechanism is integrated inherently, these models can predict and explain simultaneously with a single forward propagation without any approximations or heuristic assumptions that decrease the faithfulness of explanations. Self-explaining methods may also improve robustness [11] and provide actionable insights for directly refining model parameters without having to calibrate the dataset [21, 22].

Based on these observations, we regard self-explaining models as providing a stronger link between humans and machine learning models, reducing misunderstanding and allowing direct control of the model based on human insights. The main challenge in achieving this new level of human-machine collaboration then becomes how to ensure self-explaining models' precision both in terms of **predictive performance** and **human precision**. Human precision refers to whether models' explanations of decision-making processes align with human decision logic. Existing approaches ensure explanations to be *easy to read*, for example, by requiring explanations to be simple and smooth in a local area [11]. However, there is little guarantee that a given explanation is a *correct* rationale for prediction according to human perception. For example, the explanation "*awesome*≥2" (i.e., the word "*awesome*" appears twice in reviews) is a good rationale for positive sentiment, while "*is*≥1" ⇒ *positive sentiment* is easy to read but unreasonable to humans. Without insurance for human precision, users may constantly find unreasonable explanations, which can significantly hamper user trust and prevent them from identifying actionable insights for model refinement. An interesting research question, then, is: how can self-explaining models generate explanations that are consistent with human decision logic?

To answer this question, we need to decide what information models obtain from humans. Collecting ground-truth labels of human decision processes for every input instance [23, 24, 25, 26, 27, 28] is expensive and limits the method's scalability. Moreover, forcing the model to make decisions exactly like humans may be unwise since it could limit its data learning capability or even learn human biases that may significantly decrease the model's performance. To address this issue, it is important that humans provide guidance at a higher level that allows the models to learn freely based on data. Accordingly, we propose two desirable properties for human precision. The first property, **global coherency**, restricts the explanation form to be consistent with human reasoning logic, thereby minimizing the probability of misinterpretation. The second property, **local coherency**, requires that each explanation naturally lead to the prediction according to human perception, thereby making explanations a correct rationale for the model output. As humans can hardly provide guidance for each explanation, it is more desirable that the models can automatically guarantee local coherency based on human guidance on global coherency.

A key to satisfying these two properties is logic rules. As shown in Fig. 1a, logic rules can have flexible forms that meet human logic and preferences, making them easy to satisfy global coherency. For example, the logical connectives can be traditional (e.g., AND, OR, and NOT) or self-defined (e.g., BEFORE). Moreover, the logic rules explicitly model whether an explanation can lead to a prediction by testing the hypothesis across the entire dataset. This ensures a meaningful relationship between explanations and predictions that leads to local coherency. Fig. 1b shows that logic rule explanations achieve even higher human precision than linear-regression-based explanations with local stability, while providing a confidence score that correlates with human precision (more details in Appendix A). Lastly, logic rules of different logical connectives correspond to a diverse set of feature interactions, providing the expressive power for good predictive performance.

This paper proposes **SELOR**, a framework for upgrading a deep model with a **S**elf-**E**xplainable version with **LO**gic rule **R**easoning capability. Our work is inspired by neuro-symbolic reasoning [29], which integrates deep learning with logic rule reasoning to inherit advantages from both. The most related works in this discipline are neural-guided search that finds a global logic program that works for (most) input-output pairs [19, 30, 31, 32], or identifies a local logic program and rule for

the given instance [33, 34, 35, 36, 37]. We adopt the latter paradigm, as global explanations for deep models usually fails to possess the same predictive power that is comparable with the deep models [14]. Existing works for generating local programs or rules have achieved promising results by effectively leveraging instance-level guidance about local programs or rules [33, 34, 35], strong external knowledge such as knowledge graphs [36, 37, 38], and a small set of predefined rules [39]. However, in our scenario, there is no instance-level guidance about the ground-truth rules, and leveraging strong knowledge such as a small set of predefined rules may introduce bias into the deep networks, as shown in our experiment results of RCN [39]. To address this, we propose a logic rule reasoning framework that leverages global level human priors about rules (e.g., desirable form and property of candidate atoms) and generate explanations by optimizing rule confidence, which can be automatically computed based on the training data. Moreover, we design a neural consequent estimator that can accurately approximate the confidence even for rare rules and combine it with recursive Gumbel-Softmax [40] to search the solution space effectively. Codes are released at Github.[3]

Our main contributions are as follows.

- Our work suggests that human precision is key for self-explaining models to bridge human logic and model decision logic seamlessly. Logic rule-based explanations enable high human precision while allowing the expressive power to achieve high prediction performance.
- We propose a logic rule reasoning framework that upgrades a given deep model into a self-explainable version by naturally integrating human priors, rule confidence modeling, and rule generation as an essential part of model prediction. Our method can achieve high human precision without depending on strong external knowledge, such as instance-level guidance about rules, knowledge graphs, or a small number of rule candidates.
- Numerical experiments and user studies confirm key strengths of our framework in terms of human precision and robustness against noisy labels with maintenance of prediction performance.

## 2 Deep Logic Rule Reasoning

### 2.1 Formulation of Logic Rules

A logic rule $\boldsymbol{\alpha} \Rightarrow y$, as shown in Fig. 1a, consists of an antecedent $\boldsymbol{\alpha}$ and a consequent $y$. Meanings of symbols used in this paper are defined in Appendix B.1.

- An **antecedent** $\boldsymbol{\alpha}$ is the condition to apply the rule and corresponds to an explanation in a logic form. It is represented as a sequence $\boldsymbol{\alpha} = (o_1..., o_L)$, where $o_i$ is either an atom or a logical connective.
  - An **atom** is the smallest unit of explanation that corresponds to a single *interpretable feature* of a given input (e.g., "*awesome$\geq$2*"). The interpretable features may be different from those in deep learning models. They could, for example, have a different granularity (e.g., words or phrases) than the model features (e.g., partial words), be a statistical feature (e.g., word frequency), or be derived using external tools (e.g., grammatical tagging of a word). Mathematically, each atom $o_i$ is a Boolean-value function, with $o_i(\mathbf{x})$ returning true if the $i$-th interpretable feature is present in input $\mathbf{x}$ and false, otherwise. More detail about atom selection is in Appendix C.2
  - A **logical connective** combines atoms to form an explanation. Logical connectives can be traditional ones like AND, OR, NOT, or self-defined ones, as long as they take one or more Boolean values as the input and output a single Boolean value.
  We say that an input sample $\mathbf{x}$ **satisfies** an antecedent $\boldsymbol{\alpha}$, if $\boldsymbol{\alpha}(\mathbf{x})$ is true.
- The **consequent** $y$ is the model's prediction output given the antecedent. For example, $y$ is the predicted class in a classification task, whereas $y$ is an explicit number in a regression task. We mainly consider classification in the paper and extend the cases to regression in Appendix B.2.

### 2.2 Framework for Deep Logic Rule Reasoning

Let us denote $f$ as a deep learning model that estimates probability $p(y|\mathbf{x})$, where $\mathbf{x}$ is the input data sample and $y$ is a candidate class. We upgrade model $f$ to a self-explaining version by adding a latent variable $\boldsymbol{\alpha}$, which is an explanation in the logic form. Then, we can reformulate $p(y|\mathbf{x})$ as

$$p(y|\mathbf{x}, b) = \sum_{\boldsymbol{\alpha}} p(y|\boldsymbol{\alpha}, \mathbf{x}, b)p(\boldsymbol{\alpha}|\mathbf{x}, b) = \sum_{\boldsymbol{\alpha}} p(y|\boldsymbol{\alpha})p(\boldsymbol{\alpha}|\mathbf{x}, b), \quad s.t., \quad \Omega(\boldsymbol{\alpha}) \leq S \qquad (1)$$

---

[3]https://github.com/archon159/SELOR

Here, $b$ represents a human's prior belief about the rules, e.g., the desirable form of atoms and logical connectives, $\Omega(\boldsymbol{\alpha})$ is the required number of logic rules to explain given input $\mathbf{x}$, and $S$ is the number of samples (logic rules chosen by the model). Eq. (1) includes two constraints essential for ensuring explainability. The first constraint $p(y|\boldsymbol{\alpha}, \mathbf{x}, b) = p(y|\boldsymbol{\alpha})$ requires that explanation $\boldsymbol{\alpha}$ contains all information in the input $\mathbf{x}$ and $b$ that is useful to predict $y$. Without the constraint, the model may "cheat" by predicting $y$ directly from the input instead of using the explanation (more details in Appendix B.3). The second constraint $\Omega(\boldsymbol{\alpha}) \leq S$ requires that the model can be well explained by using only $S$ explanations, where $S$ is small enough to ensure readability ($S = 1$ in our implementation).

We can further decompose Eq. (1) based on the independence between the input $\mathbf{x}$ and the human prior belief $b$, following the generative process in Fig. 1c (proof and assumptions in Appendix B.3):

$$p(y|\mathbf{x}, b) = \sum_{\boldsymbol{\alpha}} p(y|\boldsymbol{\alpha})p(\boldsymbol{\alpha}|\mathbf{x}, b) \propto \sum_{\boldsymbol{\alpha}} \underbrace{p(b|\boldsymbol{\alpha})}_{\substack{\text{Human} \\ \text{prior}}} \cdot \underbrace{p(y|\boldsymbol{\alpha})}_{\substack{\text{Consequent} \\ \text{estimation}}} \cdot \underbrace{p(\boldsymbol{\alpha}|\mathbf{x})}_{\substack{\text{Deep antecedent} \\ \text{generation}}}, \quad s.t., \quad \Omega(\boldsymbol{\alpha}) \leq S \quad (2)$$

The three derived terms correspond to three main modules of the proposed framework, SELOR:

- **Human prior** $p(b|\boldsymbol{\alpha})$ specifies human guidance regarding desirable forms for rules to minimize the probability of misunderstanding and ensure global coherency (Sec. 2.3).
- **Consequent estimation** $p(y|\boldsymbol{\alpha})$ ensures a meaningful and consistent relationship between the explanation $\boldsymbol{\alpha}$ and prediction $y$, so that each explanation naturally leads to the prediction according to human perception and satisfies local coherency (Sec. 2.4).
- **Deep antecedent generation** $p(\boldsymbol{\alpha}|\mathbf{x})$ uses the deep representation of input $\mathbf{x}$ learned by the given deep model $f$ to find an explanation $\boldsymbol{\alpha}$ that maximizes global and local coherency (Sec. 2.5).

The sparsity constraint $\Omega(\boldsymbol{\alpha}) \leq S$ for the explanations can be enforced by sampling from $p(\boldsymbol{\alpha}|\mathbf{x})$. In particular, we rewrite Eq. (2) as an expectation and estimate it through sampling:

$$p(y|\mathbf{x}, b) \propto \sum_{\boldsymbol{\alpha}} p(b|\boldsymbol{\alpha}) \, p(y|\boldsymbol{\alpha}) \, p(\boldsymbol{\alpha}|\mathbf{x}) = \underset{\substack{\boldsymbol{\alpha} \sim \\ p(\boldsymbol{\alpha}|\mathbf{x})}}{\mathbb{E}} \, p(b|\boldsymbol{\alpha})p(y|\boldsymbol{\alpha}) \approx \frac{1}{S} \sum_{\substack{s \in [1, S] \\ \boldsymbol{\alpha}^{(s)} \sim p(\boldsymbol{\alpha}|\mathbf{x})}} p(b|\boldsymbol{\alpha}^{(s)}) \, p(y|\boldsymbol{\alpha}^{(s)}) \quad (3)$$

where $\boldsymbol{\alpha}^{(s)}$ is the $s$-th sample of $\boldsymbol{\alpha}$. For example, to maximize the approximation term with $S = 1$, the explanation generator $p(\boldsymbol{\alpha}|x)$ must find a single sample $\boldsymbol{\alpha}^{(s)}$ that yields the largest $p(b|\boldsymbol{\alpha}^{(s)})p(y|\boldsymbol{\alpha}^{(s)})$, and it needs to assign a high probability to the best $\boldsymbol{\alpha}^{(s)}$. Otherwise, other samples with a lower $p(b|\boldsymbol{\alpha}^{(s)})p(y|\boldsymbol{\alpha}^{(s)})$ may be generated, thereby decreasing $p(y|\mathbf{x}, b)$. This ensures the sparsity of $p(\boldsymbol{\alpha}|\mathbf{x})$ and the model interpretability. If there are multiple best explanations that result in the exact same $p(b|\boldsymbol{\alpha}^{(s)})p(y|\boldsymbol{\alpha}^{(s)})$, the explanation generator may find all of them.

## 2.3 Human Prior $p(b|\boldsymbol{\alpha})$

Human prior $p(b|\boldsymbol{\alpha}) = p_h(b|\boldsymbol{\alpha})p_s(b|\boldsymbol{\alpha})$ consists of hard priors $p_h(b|\boldsymbol{\alpha})$ and soft ones $p_s(b|\boldsymbol{\alpha})$.

**Hard priors** categorize the feasible solution space for the rules: $p_h(b|\boldsymbol{\alpha}) = 0$ if $\boldsymbol{\alpha}$ is not a feasible solution. Humans can easily define hard priors by choosing the atom types, such as whether the interpretable features are words, phrases, or statistics like word frequency. The logical connectives to be considered (e.g., AND, NOT) can also be chosen, as well as the antecedent's maximum length $L$. SELOR does not require a predefined rule set. Nonetheless, we allow users to enter one if it is more desirable in some application scenarios. A large solution space increases the time cost for deep logic rule reasoning (Sec. 2.6) but also decreases the probability of introducing undesirable bias.

**Soft priors** model different levels of human preference for logic rules. For example, people may prefer shorter rules or high-coverage rules that satisfy many input samples. The energy function can parameterize such soft priors: $p_s(b|\boldsymbol{\alpha}) \propto \exp(-\mathcal{L}_b(\boldsymbol{\alpha}))$, where $\mathcal{L}_b$ is the loss function for punishing undesirable logic rules. We do not include any soft priors in our current implementation.

## 2.4 Consequent Estimation $p(y|\boldsymbol{\alpha})$

Consequent estimation ensures a meaningful and consistent relationship between an explanation $\boldsymbol{\alpha}$ and prediction $y$, so each explanation naturally leads to the prediction according to human perception. This is achieved by testing the logic rule $\boldsymbol{\alpha} \Rightarrow y$ across the entire training dataset to ensure that it represents a global pattern that is typically consistent with human understanding.

**Empirical estimation**. A straightforward way to compute $p(y|\boldsymbol{\alpha})$ is to first obtain all samples that satisfy antecedent $\boldsymbol{\alpha}$, and then calculate the percentage of them that have label $y$ [8]. For example, given explanation $\boldsymbol{\alpha}$ ="*awesome*≥2", if we obtain all instances in which *awesome* appears more than twice and find that 90% of them have label $y = positive\ sentiment$, then $p(y|\boldsymbol{\alpha}) = 0.9$. Large $p(y|\boldsymbol{\alpha})$ corresponds to global patterns that naturally align with human perception. Mathematically, this is equivalent to approximating $p(y|\boldsymbol{\alpha})$ with the empirical probability $\hat{p}(y|\boldsymbol{\alpha})$:

$$\hat{p}(y|\boldsymbol{\alpha}) = n_{\boldsymbol{\alpha},y}/n_{\boldsymbol{\alpha}} \tag{4}$$

where $n_{\boldsymbol{\alpha},y}$ is the number of training samples that satisfy the antecedent $\boldsymbol{\alpha}$ and has the consequent $y$, and $n_{\boldsymbol{\alpha}}$ is the number of training samples that satisfy the antecedent $\boldsymbol{\alpha}$.

Directly setting $p(y|\boldsymbol{\alpha})$ to $\hat{p}(y|\boldsymbol{\alpha})$ can cause two problems. First, when $n_{\boldsymbol{\alpha}}$ is not large enough, the empirical probability $\hat{p}(y|\boldsymbol{\alpha})$ may be inaccurate, and the modeling of such uncertainty is inherently missing in this formulation. Second, computing $\hat{p}(y|\boldsymbol{\alpha})$ for every antecedent $\boldsymbol{\alpha}$ is intractable, since the number of feasible antecedents $A$ increases exponentially with antecedent length $L$.

**Neural estimation of categorical distribution**. To address the aforementioned problems, we jointly model $\hat{p}(y|\boldsymbol{\alpha})$ and the uncertainty caused by low-coverage antecedents with the categorical distribution and use a neural network to generalize to similar rules and better handle noise.

Assume that given antecedent $\boldsymbol{\alpha}$, $y$ follows a categorical distribution, with each category corresponding to a class. Then, according to the posterior predictive distribution, $y$ takes one of $K$ potential classes, and we may compute the probability of a new observation $y$ given existing observations:

$$p(y|\boldsymbol{\alpha}) = p(y|\mathcal{Y}_{\boldsymbol{\alpha}}, \beta) \approx \frac{\hat{p}(y|\boldsymbol{\alpha})n_{\boldsymbol{\alpha}} + \beta}{n_{\boldsymbol{\alpha}} + K\beta} \tag{5}$$

Here, $\mathcal{Y}_{\boldsymbol{\alpha}}$ denotes $n_{\boldsymbol{\alpha}}$ observations of class label $y$ obtained by checking the training data, and $\beta$ is the concentration hyperparameter of the categorical distribution that we automatically learn with backpropagation. Eq. (5) becomes Eq. (4) when $n_{\boldsymbol{\alpha}}$ increases to $\infty$, and becomes a uniform distribution when $n_{\boldsymbol{\alpha}}$ goes to 0. Thus, a low-coverage antecedent with a small $n_{\boldsymbol{\alpha}}$ is considered uncertain (i.e., close to uniform distribution). By optimizing Eq. (5), our method automatically balance the empirical probability $\hat{p}(y|\boldsymbol{\alpha})$ and the number of observations $n_{\boldsymbol{\alpha}}$. Probability $p(y|\boldsymbol{\alpha})$ also serves as the **confidence** score for the logic rule $\boldsymbol{\alpha} \Rightarrow y$.

We then employ a neural model to predict $\hat{p}(y|\boldsymbol{\alpha})$ and $n_{\boldsymbol{\alpha}}$ to better manage noise, generalize to similar rules, and improve efficiency. In particular, we obtain $A'$ samples of $\boldsymbol{\alpha}$ and compute $\hat{p}(y|\boldsymbol{\alpha})$ and $n_{\boldsymbol{\alpha}}$ by checking the training data. Here $A'$ is significantly smaller than the total number of feasible antecedents $A$ (Sec. 2.6). We use the multi-task learning framework in [41] to train the neural network with these samples. In particular, we minimize the loss in following equation.

$$\mathcal{L}_c = \frac{1}{2\sigma_p^2}||\hat{p}(y|\boldsymbol{\alpha}) - \tilde{p}(y|\boldsymbol{\alpha})||^2 + \frac{1}{2\sigma_n^2}||n_{\boldsymbol{\alpha}} - \tilde{n}_{\boldsymbol{\alpha}}||^2 + \log\sigma_p\sigma_n \tag{6}$$

$\tilde{p}(y|\boldsymbol{\alpha})$, $\tilde{n}_{\boldsymbol{\alpha}}$ are the predicted empirical probability and the coverage given by the neural model, and $\sigma_p$ and $\sigma_n$ are standard deviations of ground truth probability and coverage. More details for training the neural network are described in Appendix B.4 and Appendix. B.5, and effectiveness of the neural consequent estimator is shown in Appendix C.5.2

## 2.5 Deep Antecedent Generation $p(\boldsymbol{\alpha}|\mathbf{x})$

Deep antecedent generation finds explanation $\boldsymbol{\alpha}$ by reshaping the given deep model $f$. Specifically, we replace the prediction layer in $f$ with an explanation generator, so that the latent representation $\mathbf{z}$ of input $\mathbf{x}$ is mapped to an explanation, instead of directly mapping to a prediction (e.g., class label).

Given $\mathbf{z}$, which is the representation of input $\mathbf{x}$ in the last hidden layer of $f$, we generate explanation $\boldsymbol{\alpha} = (o_1..., o_L)$ with a recursive formulation to ensure that the complexity is linear with $L$ (Sec. 2.6). Formally, given $\mathbf{z}$ and $o_1, ...o_{i-1}$, we obtain $o_i$ by

$$\mathbf{h}_i = Encoder([\mathbf{z}; \mathbf{o}_1...; \mathbf{o}_{i-1}]), \quad p(o_i|\mathbf{x}, o_1..., o_{i-1}) = \frac{\mathbb{I}(o_i \in \mathcal{C}_i)\exp(\mathbf{h}_i^T\mathbf{o}_i)}{\sum_{\boldsymbol{\alpha}_i'} \mathbb{I}(\boldsymbol{\alpha}_i' \in \mathcal{C}_i)\exp(\mathbf{h}_i^T\boldsymbol{\alpha}_i')} \tag{7}$$

where $\mathbf{o}_i$ is the embedding of $o_i$ and $Encoder$ is a neural sequence encoder such as GRU [42] or Transformer [43]. $\mathbb{I}$ is the indicator function, and $\mathcal{C}_i$ is the set of candidates for $o_i$. Every candidate should

satisfy both global and local constraints. The hard priors in Sec. 2.3 provide the global constraint and ensure that $\boldsymbol{\alpha}$ has a human-defined logic form. The local constraint requires that $\mathbf{x}$ satisfies antecedent $\boldsymbol{\alpha}$. An atom "*awesome$\geq 2$*", for example, is sampled only if $\mathbf{x}$ mentions "*awesome*" more than once.

We then sample $o_i$ from $p(o_i|\mathbf{x}, o_1..., o_{i-1})$ in a differentiable way to ensure easy end-to-end training:

$$o_i = Gumbel(p(o'_i \in \mathcal{O}|\mathbf{x}, o_1..., o_{i-1})), \quad p(\boldsymbol{\alpha}|\mathbf{x}) = \prod_{i \in [1,L]} p(o_i|\mathbf{x}, o_1..., o_{i-1}) \quad (8)$$

$Gumbel$ is Straight-Through Gumbel-Softmax [40], a differentiable function for sampling discrete values. $o_i$ is represented as a one-hot vector with a dimension of $|\mathcal{O}|$ and is multiplied with the embedding matrix of atoms and logical connectives to derive the embedding $\mathbf{o}_i$.

### 2.6 Optimization and Complexity Analysis

**Optimization**. A deep logic rule reasoning model is learned in two steps. The first step optimizes the neural consequent estimator by minimizing loss $\mathcal{L}_c$ in Eq. (6). The neural consequent estimator only needs to be trained once for each dataset, and then it can be used for various deep models and hyperparameters. The second step converts deep model $f$ to an explainable version by maximizing $p(y|\mathbf{x}, b)$ in Eq. (3) with a cross-entropy loss. This is equivalent to minimizing loss $\mathcal{L}_d = -\mathcal{L}_b(\boldsymbol{\alpha}^{(s)}) - \log p(y^*|\boldsymbol{\alpha}^{(s)})$, where $-\mathcal{L}_b(\boldsymbol{\alpha}^{(s)})$ punishes explanations that do not fit human's prior preference for rules (global coherency), and $\log p(y^*|\boldsymbol{\alpha}^{(s)})$ finds explanation $\boldsymbol{\alpha}^{(s)}$ that leads to the ground-truth class $y^*$ with a large confidence (prediction accuracy), in which the confidence is measured by testing rule $\boldsymbol{\alpha}^{(s)} \Rightarrow y^*$ in all training data (local coherency).

**Complexity analysis**. Time complexity is compared in Table 1. The complexity for antecedent generation corresponds to the time added for generating the antecedents during model training compared to the time required for training the base deep model $f$. Here, $N$ is the number of training samples, and $C$ is the time complexity for computing the consequent of each antecedent. As shown in the table, removing the recursive antecedent generator (RG) or the neural consequent estimator (NE) brings an additional linear complexity with the number of feasible antecedents $A$, which is much larger than $A'$. For example, in our experiment, setting $A'$ to $10^4$ is good enough to train an accurate neural consequent estimator, while the number of all possible antecedents is $A = 6.25 \times 10^{12}$. Here, we do not include the analysis for sampling $A'$ rules before training the consequent estimator. See Appendix B.4 for more details.

Table 1: Time complexity analysis. -RG and -NE denote our method without recursive antecedent generation and neural consequent estimator.

| | Consequent Estimator | Antecedent Generator |
|---|---|---|
| SELOR | $O(A'C + A'L^2)$ | $O(N|\mathcal{O}|L + NL^2)$ |
| -RG | $O(A'C + A'L^2)$ | $O(NA + NL^2)$ |
| -RG-NE | $O(AC)$ | $O(NA)$ |

## 3 Experiment

### 3.1 Experimental Settings

**Datasets**. We conduct experiments on three datasets. The first two are textual, and the third is tabular. **Yelp** classifies reviews of local businesses into positive or negative sentiment [44], and **Clickbait News Detection** from Kaggle labels whether a news article is a clickbait [45]. **Adult** from the UCI machine learning repository [46], is an imbalanced tabular dataset that provides labels about whether the annual income of an adult is more than $50K/yr or not. For Yelp, we use a down-sampled subset (10%) for training, as per existing work [39]. More details about the datasets are in Appendix C.1.

**Baselines**. We compare our model to four baselines. Two self-explainable models, **SENN** [11] and **RCN** [39], are compared in accuracy, robustness, explainability, and efficiency. Two post-hoc explainable methods, **LIME** [1]and **Anchor** [14], are compared in explainability and efficiency.

**Implementation details**. To match with baselines, we use the AND operation by default in explanations. The impact of using other logical connectives is presented in Appendix C.5.3. The atoms, or interpretable features, are the same as in the majority of baselines, i.e., the existence of words for the textual dataset (e.g., "*amazing*"), and categorical and numerical features for the tabular data (e.g.,"*age$<28$*"). More details including selection of atom candidates are in Appendix C.2.

Table 2: Comparison of classification performance measured in AUC. The average results from five runs are shown. "Base" refers to the performance of unexplainable vanilla backbones. The best results among self-explaining models are marked in **bold**, and the highlighted cells indicate a similar or better result compared with the unexplainable base model. The numbers in subscript indicates the standard error of the result.

| | Yelp | | Clickbait | | Adult | Average |
|---|---|---|---|---|---|---|
| | BERT | RoBERTa | BERT | RoBERTa | DNN | |
| Base | $97.39_{0.0659}$ | $97.90_{0.0577}$ | $62.27_{1.0400}$ | $63.72_{0.8722}$ | $68.62_{0.2317}$ | 77.98 |
| SENN | $96.00_{0.1087}$ | $96.97_{0.0841}$ | $55.64_{1.0118}$ | $57.93_{0.7779}$ | $67.39_{0.0854}$ | 63.20 |
| RCN | $\mathbf{97.31}_{0.0274}$ | $\mathbf{98.03}_{0.0086}$ | $59.91_{0.2024}$ | $59.37_{0.2259}$ | $70.06_{0.0411}$ | 76.94 |
| SELOR | $97.28_{0.0335}$ | $97.78_{0.0833}$ | $\mathbf{60.31}_{0.8498}$ | $\mathbf{64.14}_{0.5906}$ | $70.36_{0.0892}$ | **77.97** |

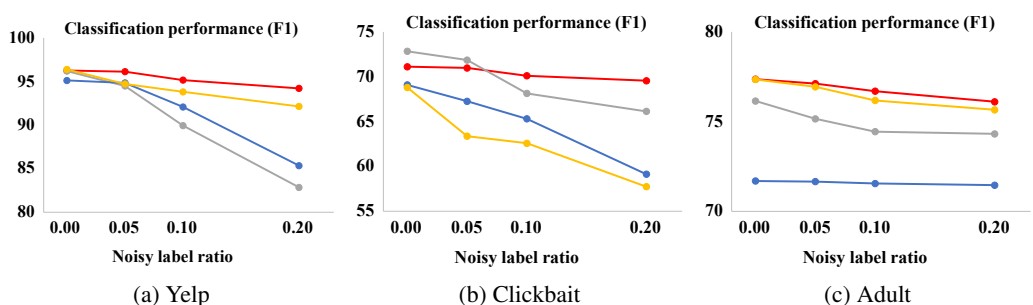

(a) Yelp  (b) Clickbait  (c) Adult

Figure 2: Experimental results on robustness under different ratios of noisy labels.

## 3.2 Classification Performance and Robustness

**Classification performance**. Table 2 shows the classification performance of SELOR and baselines. Here, we evaluate the PR AUC instead of the ROC AUC because the latter is less suitable for imbalanced datasets [47]. BERT [48] and RoBERTa [49] are used as the backbone networks in the NLP datasets, while 3-Layer DNN is used for the tabular dataset. The base method is the vanilla backbone network that does not support explainability (Appendix C.2). The prediction performance of post-hoc methods, LIME and Anchor, is the same as the base model as they utilize the trained model without any extra optimization. Comparison with a fully-transparent model is presented in Appendix C.5.1. Our method achieves comparable average performance with the unexplainable base model and outperforms other self-explaining models by 1.3%. Moreover, our method achieves the best or comparable results on various datasets against backbone models, demonstrating the expressive power of logic rules for high prediction performance. RCN cannot perform as well on challenging textual datasets like Clickbait because it computes soft attention over a predefined rule set. This indicates that (potentially biased) predefined rule sets will limit the model's capability.

**Robustness to noisy labels**. Following the literature [50, 51], we assess the robustness of SELOR against randomly corrupted labels. We hypothesize that the effect of the noisy label is alleviated by consequent estimation term $p(y|\boldsymbol{\alpha})$, where model verifies its decision by testing the logic rule over the entire dataset. For experiments, symmetric noise is introduced by randomly flipping the labels for a subset of the training data. Fig. 2 shows the results over Yelp, Clickbait, and Adult datasets with multiple levels of noise ratio from 5% to 20%. Our model outperforms other models across all scenarios. The improvement is substantial even with a high noise ratio (i.e., 20%). For a noise ratio above 10%, our method consistently outperforms the unexplainable base models (2.4% to 13.7%).

**Sensitivity analysis.** Due to space limitations, we show that the prediction performance of SELOR is stable under different hyper-parameter settings in Appendix C.5.4.

## 3.3 Explainability

**User study on human precision**. To evaluate human precision, we recruited nine native English speakers through a vendor company [52]. Each participant was provided with randomly selected 50 Yelp reviews and 50 Adult samples. Five explanations obtained from different methods were provided for each sample, and the participants reviewed whether the explanations offered reasonable rationales. Participants provided two labels for each explanation, indicating whether it was good

Table 3: User study results on human precision. We show the average (Avg.) and inter-participant agreement (Agr.) on the percentage of explanations that are considered good (a, b) or best (c, d). One star (*) means p-value is less than 0.05. Best results are highlighted in **bold**.

<table>
<tr><td colspan="4">(a) Percentage of good (Yelp)</td><td colspan="4">(b) Percentage of good (Adult)</td></tr>
<tr><td></td><td>Avg.</td><td>Agr.</td><td>P-value</td><td></td><td>Avg.</td><td>Agr.</td><td>P-value</td></tr>
<tr><td>Lime</td><td>89.8</td><td>84.4</td><td>8.68 E-04*</td><td>Lime</td><td>42.7</td><td>57.1</td><td>6.09 E-54*</td></tr>
<tr><td>Anchor</td><td>84.4</td><td>87.7</td><td>1.12 E-07*</td><td>Anchor</td><td>52.7</td><td>59.9</td><td>5.56 E-18*</td></tr>
<tr><td>SENN</td><td>34.4</td><td>72.3</td><td>1.40 E-51*</td><td>SENN</td><td>46.0</td><td>51.5</td><td>1.18 E-41*</td></tr>
<tr><td>RCN</td><td>64.0</td><td>77.6</td><td>7.26 E-13*</td><td>RCN</td><td>60.9</td><td>53.2</td><td>2.83 E-27*</td></tr>
<tr><td>SELOR</td><td>**94.4**</td><td>93.9</td><td>-</td><td>SELOR</td><td>**90.7**</td><td>85.7</td><td>-</td></tr>
<tr><td colspan="4">(c) Percentage of best (Yelp)</td><td colspan="4">(d) Percentage of best (Adult)</td></tr>
<tr><td></td><td>Avg.</td><td>Agr.</td><td>P-value</td><td></td><td>Avg.</td><td>Agr.</td><td>P-value</td></tr>
<tr><td>Lime</td><td>34.2</td><td>67.6</td><td>8.87 E-03*</td><td>Lime</td><td>1.3</td><td>96.7</td><td>1.72 E-64*</td></tr>
<tr><td>Anchor</td><td>18.0</td><td>83.6</td><td>5.63 E-18*</td><td>Anchor</td><td>13.8</td><td>82.9</td><td>1.23 E-35*</td></tr>
<tr><td>SENN</td><td>2.4</td><td>96.3</td><td>6.84 E-40*</td><td>SENN</td><td>9.6</td><td>83.3</td><td>2.30 E-36*</td></tr>
<tr><td>RCN</td><td>2.0</td><td>96.3</td><td>6.84 E-40*</td><td>RCN</td><td>10.2</td><td>82.9</td><td>1.23 E-35*</td></tr>
<tr><td>SELOR</td><td>**46.7**</td><td>64.8</td><td>-</td><td>SELOR</td><td>**65.1**</td><td>58.4</td><td>-</td></tr>
</table>

or if it was the best one. A **good** explanation should naturally lead to the prediction, but it can contain noisy features. For example, "*amazing*, *are*" is a good explanation for positive sentiment. The **best** explanation is the one that contains the most important and least noisy features. The participants were allowed to choose multiple best explanations only if the chosen ones were the same. For a fair comparison, we showed explanations in the same form: a list of features each method considers important for prediction. Example explanations generated by our method and the baselines are shown in Fig. 3. Note that ˜ in RCN means negation. More details about the explanation generation, labeling guidelines, and participants' results are given in Appendix C.4.

Table 3 shows that SELOR marks the highest percentage of good explanations, with an average ratio of 94.4% on Yelp and 90.7% on Adult. Our method is also most frequently chosen as the best explanation. All results are statistically significant according to the p-values from the t-tests. Although logic rules are promising, choosing from a small set of predefined rules may be insufficient due to the potential bias in the rule set. For example, RCN uses rules extracted with traditional machine learning methods that meet the global data distribution but frequently fail to adequately represent each sample, particularly on datasets with many features like Yelp. As a result, RCN is rarely chosen as the best explanation, especially for Yelp text data. Post-hoc methods also tend to offer good human precision. The best ratio of LIME and Anchor, however, is substantially lower than ours, indicating that the base model may rely on more noisy features for prediction. In contrast, our method can verify its decision by testing the logic rule across the entire dataset.

**Case study on model debugging and refinement**. What useful insights can SELOR provide on performance? In a study of 20,000 sampled Yelp reviews, we clustered the generated explanations into 10 clusters by applying K-Means on the antecedent embeddings. Table 4 shows five clusters with the lowest training accuracy to illustrate **potential reasons** for bad performance. Here, NULL is an empty atom when the model generated explanations that were shorter than the predefined length $L$.

We make the following observations. First, low training accuracy in **cluster 5** is due to non-English reviews, which accounted for 0.31% and led to underfitting. Second, performance degradation also

*…The staff was amazing with him and made him feel so comfortable that he actually sat through an entire cleaning without being upset once…The dentist and the assistants were made to work with children and that made it all very relaxing and enjoyable for him.*

| | |
|---|---|
| LIME | amazing, comfortable, enjoyable, more |
| Anchor | amazing, so |
| SENN | about |
| RCN | amazing, ˜went, ˜seeing, ˜did |
| Ours | amazing, relaxing, comfortable, enjoyable |

| Adult | | LIME | Anchor | SENN | RCN | Ours |
|---|---|---|---|---|---|---|
| age | 27 | | | | <32.5 | <28 |
| education | HS-grad | ✓ | | | | |
| educational-num | 9 | | | | <12.5 | <10 |
| marital-status | Separated | | ✓ | ✓ | ✓ | |
| occupation | Craft-repair | | | ✓ | | ✓ |
| relationship | Own-child | ✓ | | | ✓ | |
| race | White | | | | | |
| capital-gain | 0 | ✓ | | ✓ | | |
| capital-loss | 0 | ✓ | | ✓ | | |

Figure 3: Example explanations produced by five methods on Yelp (left) and Adult (right).

Table 4: Case study on Yelp. We cluster the explanations for the training samples and show the five clusters with the lowest training accuracy. Num is the number of explanations in the cluster, and Len is the average text length of the reviews. Potential reasons for bad performance are marked in **brown**.

| Cluster | Acc | Label | Num | Len | Atoms in the explanations (ordered by frequency) |
|---|---|---|---|---|---|
| 1 | 99.2 | 99.1% Neg | 1,763 8.82% | 643 | not(290) bad(233) no(185) mediocre(153) bland(149) never(123) again(122) worst(119) ok(115) disappointing(115) terrible(107) |
| 2 | 99.2 | 99.2% Pos | 2,730 13.7% | 584 | great(667) delicious(508) best(330) love(294) fresh(285) tasty(255) definitely(254) friendly(232) perfect(199) amazing(198) favorite(194) |
| 3 | 98.6 | 98.6% Pos | 2,686 13.4% | 548 | great(793) friendly(329) always(315) best(304) love(295) fun(223) definitely(218) helpful(163) awesome(159) amazing(136) **vegas(117)** |
| 4 | 93.2 | 57.8% Pos | 848 4.24% | **119** | **NULL(1738)** great(133) not(67) best(39) service(39) love(34) friendly(29) good(26) awesome(22) fast(22) overpriced(20) |
| 5 | 83.9 | 58.1% Neg | 62 **0.31%** | 682 | **NULL(24) nicht(13) un(11) eine(9)** service(8) **pas(8) der(8)** die(7) **und(7) um(6) den(5) pour(5) de(5) das(4) prix(4) je(4) zu(3) im(3)** |

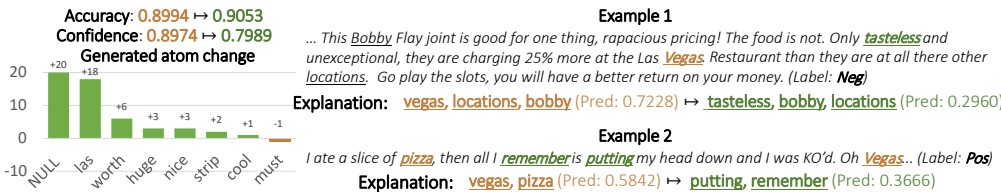

Figure 4: Steering the model without re-training. SELOR allows users to exclude noisy features from explanations during testing, which may simultaneously improve the explanation quality and prediction accuracy. This figure shows the performance **before** and **after** removing "*vegas*".

happens when the model does not have enough evidence. For example, reviews in **cluster 4** were short (average length of 119 words) and contained an overwhelming number of NULL atoms (on average 2 per explanation). Third, **cluster 3** contained 13.4% samples with positive sentiment, and its training accuracy (98.6%) is higher than cluster 4. However, the cluster often included "*vegas*" in the explanation, which does not seem directly related to sentiment classification. Fourth, **clusters 1 and 2** have reasonable atoms, which seem consistent with high training accuracy (99.2%).

SELOR allows us to steer the model directly. For example, after identifying the potentially noisy feature like "*vegas*", we can prevent the model from including the term by removing it from the candidate atom list $\mathcal{C}_i$. This type of refinement can be easily achieved during testing, unlike the efforts-taking dataset calibration or model retraining. Fig. 4 shows the performance change of the 169 samples that previously included "*vegas*" in their explanations. The histogram shows which atoms that are generated more after removing "*vegas*". The model sometimes relies on similar atoms such as "*las*" or does not find a good candidate (e.g., choosing NULL), which may lead to decreased confidence. However, the chance of including more meaningful atoms also increases (e.g., "*worth*" in the histogram, and "*tasteless*" in Example 1). One may also verify assumptions by checking the samples whose prediction score changes. For instance, after removing "*vegas*", the model can no longer predict Example 2 correctly. The example contains no obvious indication of sentiment, and "*vegas*" may be the most helpful feature. This contradicts our previous assumption that "*vegas*" seems not critical for sentiment classification. Instead it can provide new insights and guidance for further improvement (e.g., punishing "*vegas*" with a soft prior instead of directly removing it).

**Explanation stability and sensitivity analysis**. We discuss the stability of our explanations in Appendix C.6. Our quantitative experiment demonstrates that the explanations generated in different runs are consistent. We also present a case study in that SELOR gives similar explanations for similar inputs. Moreover, we discuss user study results that the human precision of the explanations is good across different hyper-parameter settings in Appendix C.5.4.

## 3.4 Efficiency

Table 5 shows that post-hoc explanation methods like LIME and Anchor require a longer time to generate an explanation. RCN has the largest complexity among the self-explaining methods since

Table 5: Time costs in seconds on Yelp (BERT) and Adult.

| | Consequent estimator training | Deep model training (1 epoch) | | | | Explanation generation (1 sample) | | | | |
|---|---|---|---|---|---|---|---|---|---|---|
| | | Base | SENN | RCN | SELOR | LIME | Anchor | SENN | RCN | SELOR |
| Yelp | 2041.4 | 571.9 | 224.5 | 503.7 | 665.6 | 55.0 | 2854.2 | 0.037 | 0.071 | 0.055 |
| Adult | 1502.8 | 12.8 | 9.1 | 953.9 | 98.2 | 2.5 | 1.18 | 0.02 | 0.17 | 0.015 |

it enumerates all possible rules and combines them with soft attention. To alleviate this problem, RCN uses a predefined rule set; hence, its efficiency becomes dependent on the size and quality of the rule set. In contrast, SELOR is trained within acceptable time even for large solution space, and humans only need to define the types of atoms and logical connectives. Our model generated each explanation with a linear complexity with length $L$, while RCN goes over all possible rules and has exponential complexity with $L$. Our method required additional time for the neural consequent estimator, taking 35 minutes on Yelp and 25 minutes on Adult. This step is only required once for each dataset and hence is acceptable. The consequent estimator can also be reused.

## 4  Conclusion and Future Work

This work presented a new framework, SELOR, which incorporates self-explanatory capabilities into a deep model to provide high human precision by explaining logic rules while also maintaining high prediction performance. Our method does not require predefined rule sets and can be learned in a differentiable way. Extensive tests involving human evaluation show that our method achieves high prediction performance and human precision while being resistant to noisy labels. Although our method brings substantial advantages, there remain multiple aspects for improvement in the future:

**Stability**. A desirable property for self-explaining models is stability, which requires that similar inputs lead to similar explanations. Unlike SENN [11], which proposes a robustness loss to ensure stable explanations against adversarial inputs, SELOR does not employ such a constraint and cannot guarantee the stability of explanations for inputs with similar raw features. However, our framework theoretically ensures stability is modeled in the selected feature space (see Appendix B.6 for more details), which is partially evaluated by a case study in Appendix C.6.

**Applicability**. While we explored text and tabular data, our model is applicable to other data types like images and graphs. We can treat a cluster of images or superpixels as an atom [11] or extract atoms with CAV (Concept Activation Vector), a feature that indicates the concept of humans (e.g., striped, red) [53]. End-to-end feature learning is possible in our framework if the number of candidate atoms is small (e.g., around 100 object classes or concepts [35]).

**Level of insight**. SELOR cannot explicitly model higher-level properties of atoms (e.g., learn that "*awesome*" is a *positive sentiment word* and make a rule based on *positive sentiment word*) since we do not directly consider predicates. We can only find rules constructed with bottom-level atoms instead of summarizing important high-level patterns, which also leads low coverage of rules (e.g., rule "*awesome* AND *tasty*" => *positive sentiment* only covers 0.37% of the input instances). If re-designed as the first-order logic, the model may directly find high-level patterns such as "*a negation word* AND *a positive sentiment word*" => *negative sentiment*, instead of listing many specific rules such as "*not great*" => *negative sentiment* and "*no good*" => *negative sentiment*. This could save human cognitive budget and improve the reasoning capability of deep models. Moreover, we may automatically compose high-level concepts such as "*strong positive phrase*" and build rules with them. The concept "*strong positive phrase*" may be composed by detecting two consecutive positive sentiment words ("*amazingly comfortable*" and "*perfectly enjoyable*") with predicate invention in [19].

## Acknowledgments and Disclosure of Funding

We thank Fangzhao Wu, Sundong Kim, and Eunji Lee for their insightful feedback on our work. We appreciate the reviewers of this paper for their valuable suggestions that improved the paper significantly. This research was supported by Microsoft Research Asia, the Institute for Basic Science (IBS-R029-C2) in Korea, and the Potential Individuals Global Training Program (2021-0-01696) by the Ministry of Science and ICT in Korea.

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
