# OpenReview forum: "Self-explaining deep models with logic rule reasoning"
_NeurIPS.cc/2022/Conference — NeurIPS 2022 Accept_

### Official Review · Reviewer_pby1 · 2022-07-11

**Rating:** 6
**Confidence:** 3
**Soundness:** 2 fair
**Presentation:** 3 good
**Contribution:** 3 good

**Summary:**

The paper proposes an interpretable-by-design framework for deep learning, leveraging logical rules in the form of propositional implications to explain model’s predictions.
Specifically, the body of the rule (alternatively called antecedent in the article) is predicted by a carefully chosen deep learning architecture, whereas the head (alternatively called consequent) consists of the result of the implication rule, namely the label prediction. The deep learning component includes (i) a backbone encoding network to learn a representation from the raw input data and (ii) an autoregressive model transforming the latent representation into the predicting atoms (Boolean features) composing the body of the logical rule.
The training is performed in two stages. In particular, an auxiliary network is first trained to mimic the prediction and the confidence of random implication rules. The learnt network is then used as a prediction head for the above mentioned deep learning component to minimise the prediction error together with additional user specific constraints in an end-to-end differentiable fashion.

The proposed approach is compared against  two self-explainable and two post-hoc explanation solutions on two textual and one tabular datasets. The results show that the proposed model is able to outperform previous approaches in terms of explainability as measured by human precision, while retaining state-of-the-art predictive performance.


**Questions:**

1. Can you please elaborate on the novelty of the proposed framework in relation to neuro-symbolic approaches?
2. What are the soft constraints and loss used in the experiments?


**Limitations:**

1. Are there any limitations on the chosen class of logic? How would this influence the human precision?

MINOR MISTAKE

Please replace $p(\alpha|b)$ with $p(b|\alpha)$ in Eq. (1), line 115, line 131, multiple occurrences in line 132, line 133, line 142 and Eq. (6). Also, modify the generative process in Figure 1 (c) to take into account the correct dependence between $\alpha$ and $b$

--------------------------------------------------
POST - REBUTTAL

During the rebuttal, the authors have properly addressed all my questions. Also, they have provided concrete actions to take into account the weaknesses of the manuscript (i.e. missing related work on neuro-symbolic learning, discussion on language expressiveness). Overall, the work provides an interesting contribution for the application of neuro-symbolic learning in the area of explainable AI. I'm happy to recommend for a weak accept.


**Strengths And Weaknesses:**

**Strengths**

- The paper is clearly written and the use of logical rules is well motivated.
- The considered problem is relevant and timely.
- Experiments show the superiority of the proposed solution over the baselines especially in terms of explainability. I liked the analysis of the predictions in Section 3.3, as providing convincing evidence on the utility of logical rules. This is an original part of the work.

**Weaknesses**

- The paper overlooks an important body of research in the emerging field of neuro-symbolic learning. Indeed, this is not the first time that rules are learnt in symbiosis with deep learning, see for example [1] and [2]. Experimental comparisons would be required against these strategies, but I realise that at this stage this could not be feasible. Authors should at least cite and discuss works in neuro-symbolic learning and highlight the point of novelties of their proposed solution.
- The work focuses on learning logical rules in the form of definite clause logic (a subclass of propositional logic), thus inheriting its limitations in terms of expressivity and also explainability. A discussion on these limitations is required.

[1] Learning Explanatory Rule From Noisy Data. JAIR 2018
[2] DeepProbLog: Neural Probabilistic Logic Programming. NeurIPS 2018

---

> ### Author Response · Authors · 2022-08-02
> **Response to Reviewer pby1 [1/3]**
>
> > Comparison with neuro-symbolic approaches
> > - **[Weakness 1]** The paper overlooks an important body of research in the emerging field of neuro-symbolic learning. Indeed, this is not the first time that rules are learnt in symbiosis with deep learning, see for example [1] and [2]. Experimental comparisons would be required against these strategies, but I realise that at this stage this could not be feasible. Authors should at least cite and discuss works in neuro-symbolic learning and highlight the point of novelties of their proposed solution.
> > - **[Question 1]** Can you please elaborate on the novelty of the proposed framework in relation to neuro-symbolic approaches?
>
> We appreciate the reviewer's suggestions for related research to help us better position our work in the literature. The suggested methods [1][2] cannot be directly applied here due to the scalability issue mentioned as follows. Moreover, we have additional contributions related to quantification of explainability and easy to use. We will carefully discuss these in the revised paper.
>
> It is true that we are not the first to integrate deep learning with logic rule reasoning. Instead, we consider our research as the **first to quantify the importance of logic rules in self-explainable AI** (with human precision). We present logic rules as an essential and indispensable explanation type that the explainable AI community has long overlooked, rather than as an optional option. To achieve this goal, we revisited the explainability literature, argued that human precision is the missing link to seamlessly bridging human and model decisions, and demonstrated that logic rule explanations bridge the gap by naturally achieving much greater human precision. We anticipate that this will draw attention from the explainable AI community to the critical field of neuro-symbolic reasoning.
>
> In addition, our paper contributes **a novel reasoning framework** that solves two major challenges when explaining a deep model by using logic rules.
>
> The first challenge our method solves is the **scalability** to the number of rules. Since the optimal reasoning process for a deep model in real-world scenarios is typically vague (e.g., income prediction or clickbait detection), humans cannot define the optimal rules precisely. Therefore, we searched for the optimal rules among a large number of candidate rules, e.g., the number of candidate rules is $A=6.25 × 10^{12}$ in the Yelp dataset. Most existing works in neuro-symbolic reasoning, however, either assume that the optimal rules (or logic programs) are predefined (e.g., by humans) [2] or search for the optimal rules with a complexity that is proportional to the number of rule candidates [1]. These methods are useful for tasks with a relatively straightforward reasoning procedure (e.g., computing math equations), but **they cannot be directly applied to our scenarios due to high complexity** (e.g., applying [1] to Yelp would require at least 1TB of GPU memory to store the rule weights). In contrast, our model's complexity is sublinear to the number of rule candidates (see Sec. 2.6., Complexity Analysis). This is because **we integrate deep learning differently**. While [1,2] use deep learning to extract an atom or fact from the input instance (e.g., to determine that an image depicts the number "1"), our work uses deep learning to generate the optimal rule (e.g., generate the rule that "1 AND 2 AND 3" results in "6" or "even number"). Thus, we can utilize deep models as a rule searching policy (our deep antecedent generator) and leverage neural networks to approximate the result without checking each logic rule (our consequent estimator). This significantly reduces complexity.
>
> The second challenge our method solves is **easy to use**. We hope that our framework can be easily used by explainable AI researchers who are not experts in logic reasoning. Thus, it would be ideal if our (first version) framework can be implemented easily with widely-used deep learning modules. To achieve this goal, we utilize the recent advance in deep learning (Gumbel softmax) and extend it (to recursive Gumbel softmax) so that the antecedent generator **can be learned in a differentiable way with commonly used deep learning modules**. This simplicity is partially due to the fact that we only consider definite clause logic, as you mentioned. Such a limitation may be solved by leveraging existing techniques in logic programming, e.g., by sampling the random rules for training the consequent estimator from the rule template in [1]. In addition, there are other aspects to learn from existing neuro-symbolic methods (e.g., inferring the fact or atom from images), which we believe may lead to a new line of interesting research work. These limitations and future works will be discussed in the revised paper.
>
> [1] Learning Explanatory Rule From Noisy Data. JAIR 2018
>
> [2] DeepProbLog: Neural Probabilistic Logic Programming. NeurIPS 2018

---

> > ### Comment · Reviewer_pby1 · 2022-08-04
> > **New Limitations and Possible Issues**
> >
> > Thank you for your extensive answers. While I agree with most replies, there are still some points that are unanswered. Furthermore, by reading the other reviews, additional limitations/issues have emerged. Please, find below further observations with related questions. All these points prevent me to raise the score.
> >
> > **Missing references about structure learning in symbolic strategies**
> >
> > While I appreciate the discussion about the two references and generally agree with your point, the answer is not complete. Indeed, the aim of my question was to point out that there is a large body of research aiming to perform structure learning for symbolic programs (using neural nets). For example, please refer to section 6 of [1] and specifically to neurally-guided search. This is currently overlooked in the paper. Can you please elaborate on this and explain why your solution should be considered novel in this regard?
> >
> > [1] IJCAI 2020 From Statistical Relational to Neural-Symbolic Artificial Intelligence
> >
> > **Scope of applicability and complexity of proposed solution based on neural network (why not decision trees?)**
> >
> > 1. Apart from the missing references, while reading the questions raised by reviewer oPY7, I realised that you manually engineer the candidate set used for the explanation (namely, you extract frequent words in text data, specific attribute values or quantized levels for tabular data). Indeed the neural network is only selecting elements from the candidate set and not really performing feature learning.
> >
> > 2. Furthermore, I don’t agree about the easiness of the proposed solution (transformer + autoregressive + transforme). Indeed, this is a rather complicated architecture, which brings in additional computational and optimisation issues.
> >
> > Regarding point 1. The manual design of the candidate set is indeed a limiting factor. While I agree that the tasks you provided are suited for such assumption, is it still valid for other tasks involving for example image data? This limits the scope and applicability of the proposed strategy.
> >
> > Regarding point 2. If your consider a boolean feature space over the manually defined candidate set (where the values refer to the presence or absence of elements in the data from the candidate set), then you can use a decision tree to select the most relevant elements for prediction. Importantly, it is possible to extract Boolean decision rules from the decision tree. Therefore, why should one prefer your solution over a decision tree (which is much simpler computationally speaking and also interpretable by nature)?
> >
> > Note also that this simple solution solves the issue about the self-consistency of explanations among different input samples, as raised by reviewer YUk5.
> >
> > **Technical details**
> >
> > I would recommend to add more details in the main paper/supplementary about the losses and the hyperparameters used in all your experiments. Simply, providing a high-level description of losses and architectures is not enough to provide a complete description of your solution. However, I do appreciate that you are sharing the code.

---

> > > ### Author Response · Authors · 2022-08-06
> > > **Additional Response to Reviewer pby1 [1/4]**
> > >
> > > > Missing references about structure learning in symbolic strategies
> > > > - While I appreciate the discussion about the two references and generally agree with your point, the answer is not complete. Indeed, the aim of my question was to point out that there is a large body of research aiming to perform structure learning for symbolic programs (using neural nets). For example, please refer to section 6 of [1] and specifically to neurally-guided search. This is currently overlooked in the paper. Can you please elaborate on this and explain why your solution should be considered novel in this regard?
> > > [1] IJCAI 2020 From Statistical Relational to Neural-Symbolic Artificial Intelligence
> > >
> > >  Thank you for your quick response and insightful comments. We value your expertise in neuro-symbolic reasoning and your assistance in guiding us to the most relevant works in the field. We will revised the paper based on your suggestions and make the following changes to the Introduction (lines 76-80). Additional ideas are always welcome.
> > >
> > > *“...Our work is inspired by neuro-symbolic reasoning [1,2,3,4,5,6,7,8], which integrates deep learning with logical reasoning to combine the advantages of both. The most related works in this discipline are neural guided search that finds a global logic program which works for (almost) all input-output pairs [8, 9, 10, 13], or identifies a local logic program or rule for the given instance [11,12,15,16,17]. We adopt the latter paradigm, as global explanations for deep models are usually not succinct or fail to possess the same predictive power that is comparable with the deep models [19]. Existing works for generating local programs or rules have achieved promising results by effectively leveraging instance-level guidance about ground-truth programs or rules [11,12,15] or strong external knowledge such as knowledge graphs [16, 17, 20] and a small set of predefined rules [18]. However, in our scenario, only the class labels of instances are given, and leveraging strong knowledge such as restricting the candidate rules to a small set may introduce bias into the deep networks, as shown in our experimental results of RCN [18]. To address this, we generate explanations by optimizing rule confidence, which can be automatically computed based on the training data. Moreover, we design a neural consequent estimator that can accurately approximate the confidence for unseen rules and combine it with recursive Gumbel-Softmax [21] to search the solution space effectively.”*
> > >
> > > [1] A Survey on Neural-symbolic Systems, Arxiv, 2021
> > >
> > > [2] Deepproblog: Neural probabilistic logic programming, NeurIPS, 2018
> > >
> > > [3] From Statistical Relational to Neuro-Symbolic Artificial Intelligence, IJCAI, 2020
> > >
> > > [4] Deepstochlog Neural stochastic logic programming, AAAI, 2022
> > >
> > > [5] Learning Relational Representations with Auto-encoding Logic Programs, IJCAI, 2020
> > >
> > > [6] Mapping probability word problems to executable representations, EMNLP, 2021
> > >
> > > [7] Neural Probabilistic Logic Programming in DeepProbLog, Artificial Intelligence, 2021
> > >
> > > [8] Learning Explanatory Rules from Noisy Data, Journal of Artificial Intelligence Research 61, 2018
> > >
> > > [9] HOUDINI: Lifelong Learning as Program Synthesis, NeurIPS, 2018
> > >
> > > [10] Learning Libraries of Subroutines for Neurally–Guided Bayesian Program Induction, NeurIPS, 2018
> > >
> > > [11] Learning to Infer Graphics Programs from Hand-Drawn Images, NeurIPS, 2018
> > >
> > > [12] The Neuro-Symbolic Concept Learner: Interpreting Scenes, Words, and Sentences From Natural Supervision, ICLR, 2019
> > >
> > > [13] Neural-Guided Deductive Search for Real-Time Program Synthesis from Examples, ICLR, 2018
> > >
> > > [14] A theory of consciousness from a theoretical computer science perspective: Insights from the Conscious Turing Machine, PNAS, 2022
> > >
> > > [15] A Probabilistic Graphical Model Based on Neural-symbolic Reasoning for Visual Relationship Detection, CVPR, 2022
> > >
> > > [16] Multi-level Recommendation Reasoning over Knowledge Graphs with Reinforcement Learning, The Web Conference, 2022
> > >
> > > [17] Differentiable Learning of Logical Rules for Knowledge Base Reasoning, NeurIPS, 2017
> > >
> > > [18] Deep Neural Networks Constrained by Decision Rules, AAAI, 2019
> > >
> > > [19] Anchors: High-precision model- agnostic explanations, AAAI, 2018
> > >
> > > [20] Leveraging Demonstrations for Reinforcement Recommendation Reasoning over Knowledge Graphs, SIGIR, 2020
> > >
> > > [21] Categorical Reparameterization with Gumbel-Softmax, Stat, 2017
> > >
> > > *(not finished, see the next message: response [2/4])*

---

> > > > ### Comment · Reviewer_pby1 · 2022-08-07
> > > > **Thank You, Clarification about Decision Trees**
> > > >
> > > > Thank you again for the extensive answers. I think that the contribution in terms of application of neuro-symbolic strategies to the explainibility context is clearer now. Please, find below some few but still imortant comments.
> > > >
> > > > **Related work**
> > > >
> > > > Citing one of your favourite survey on neuro-symbolic learning is more than enough, there is no need for citing several neuro-symbolic frameworks, also because the provided list is not exhaustive. Rather, it is good to focus on the distinction between different approaches for rule learning. The description you provide can be made more precise. Indeed, you mention that your proposed solution uses less information (“only labels” ) compared to existing strategies. That is not true. Your solution requires to carefully choose the candidate set in order to find good explanations. Can you please rephrase it?
> > > >
> > > > **Comparison with decision trees**
> > > >
> > > > Thanks for providing a comparison with random forest. I have only few remaining questions: Are you using the same candidate set used in your solution as the feature space for the rule learner? Are you also including the hard constraints to restrict the search?
> > > > If that is not the case, can you please explain why this should be considered a fair comparison?

---

> > > > > ### Author Response · Authors · 2022-08-09
> > > > > **Additional Response to The Reply of Reviewer pby1 [1/2]**
> > > > >
> > > > > >Related work
> > > > > > - Citing one of your favourite survey on neuro-symbolic learning is more than enough, there is no need for citing several neuro-symbolic frameworks, also because the provided list is not exhaustive. Rather, it is good to focus on the distinction between different approaches for rule learning. The description you provide can be made more precise. Indeed, you mention that your proposed solution uses less information (“only labels” ) compared to existing strategies. That is not true. Your solution requires to carefully choose the candidate set in order to find good explanations. Can you please rephrase it?
> > > > >
> > > > > Thanks for your suggestions. We have changed the references in the first sentence to a survey, replaced “only the class labels of instances are given” to “there is no instance-level guidance about the ground-truth rules”, and refined other descriptions accordingly. Please find the revised paragraph as follows, in which the changed parts are highlighted in **bold**. The numbers of the references are consistent with that in our previous response. Please feel free to let us know if there are any other issues.
> > > > >
> > > > > *“...Our work is inspired by neuro-symbolic reasoning **[3]** ... Existing works for generating local programs or rules have achieved promising results by effectively leveraging instance-level guidance about ground-truth programs or rules [11,12,15] or strong external knowledge such as knowledge graphs [16, 17, 20] and a small set of predefined rules [18]. However, in our scenario, **there is no instance-level guidance about the ground-truth rules**, and leveraging strong knowledge such as restricting the candidate rules to a small set may introduce bias into the deep networks, as shown in our experimental results of RCN [18]. To address this, we **propose a logic rule reasoning framework that leverages global level human priors about rules (e.g., desirable form and property of candidate atoms) and** generate explanations by optimizing rule confidence, which can be automatically computed based on the training data…”*

---

> > > > > > ### Comment · Reviewer_pby1 · 2022-08-09
> > > > > > **Thanks**
> > > > > >
> > > > > > Thanks for the clarifications. All my questions have been addressed and I'm happy to increase my score. Congratulations

---

> > > > > ### Author Response · Authors · 2022-08-09
> > > > > **Additional Response to The Reply of Reviewer pby1 [2/2]**
> > > > >
> > > > > >Comparison with decision trees
> > > > > > - Thanks for providing a comparison with random forest. I have only few remaining questions: Are you using the same candidate set used in your solution as the feature space for the rule learner? Are you also including the hard constraints to restrict the search? If that is not the case, can you please explain why this should be considered a fair comparison?
> > > > >
> > > > >  Thanks for the question. For our baseline RCN, we are using the suggested setting in the RCN paper [1] rather than our own setting, because our setting leads to a significant performance drop in the clickbait dataset. Here is a detailed comparison of the settings:
> > > > >
> > > > > * **RCN setting** (automatically learned atoms): use all the features, choose the split threshold (e.g., “25” in “age>25”) for numerical features according to information gain.
> > > > > * **Our setting** (manually defined atom set): use only the top 5,000 features, set the split threshold to 0.5 for words (i.e., only consider whether a word appears or not), and set the threshold to 25%, 50%, 75% percentile of data for numerical features.
> > > > > * **Hybrid setting**: use only the top 5,000 features, but choose the split threshold of words according to information gain.
> > > > >
> > > > > As you can see in table 1, our setting results in a significant performance drop in the clickbait dataset for RCN, and leads to similar results in the other datasets. It indicates that our current setting is unfair for RCN, as it sometimes limits their modeling capability and prevents them from finding the most informative words and split thresholds. Thus, we use the setting in the RCN paper for our baseline RCN. Also, please note that the hybrid setting of the tabular dataset (Adult) is the same with our setting, as there are fewer than 5,000 features in the tabular dataset.
> > > > >
> > > > > **Table 1 Performance comparison of RCN on various settings.**
> > > > > |  | Yelp |  | Clickbait |  | Adult |  |
> > > > > |---|---|---|---|---|---|---|
> > > > > |  | F1 | AUC | F1 | AUC | F1 | AUC |
> > > > > | RCN (RCN Setting) | 97.36 | 98.03 | 68.64 | 59.37 | 77.35 | 70.06 |
> > > > > | RCN (Our Setting) | 97.36 | 98.02 | 47.28 | 37.40 | 77.76 | 70.67 |
> > > > > | RCN (Hybrid Setting) | 97.36 | 98.04 | 46.94 | 34.08 | 77.76 | 70.67 |
> > > > >
> > > > > For random forests, we compute their prediction performance given the three above-mentioned settings and present their results under the best setting for each dataset. We could find that our model achieves significantly better performance to random forest in all datasets. This shows that random forests cannot achieve a comparable performance to deep learning models, regardless of which hard human prior is considered.
> > > > >
> > > > > **Table 2 Performance comparison of fully transparent model and our model.**
> > > > > |  | Yelp |  | Clickbait |  | Adult |  |
> > > > > |---|---|---|---|---|---|---|
> > > > > |  | F1 | AUC | F1 | AUC | F1 | AUC |
> > > > > | RF (Best Setting) | 76.52 | 82.41 | 50.86 | 61.48 | 70.64 | 67.63 |
> > > > > | Ours (Our Setting) | 97.13 | 97.78 | 74.2 | 64.14 | 77.37 | 70.36 |
> > > > >
> > > > > We further investigate the sensitivity of our method in terms of the hard constraints on the number of atoms used. As shown in Table 3, **our choice of using the top 5,000 features does not significantly impact model performance (p-value>0.1)**, compared with our model with all the features (about 16,000 features). This is consistent with the fact that our prediction performance and user study results do not change significantly with the number of atoms when it is larger than 1,000 (Supplement C.4.4  and Response to Reviewer zwRN [3/4]).
> > > > >
> > > > > **Table 3 Performance under different numbers of features (atoms) in Yelp**
> > > > > |  | F1 | AUC |
> > > > > |---|---|---|
> > > > > | Ours (All Features) | 97.02 | 97.71 |
> > > > > | Ours (Our Setting, 5000 Features) | 97.13 | 97.78 |
> > > > > | p-value | 0.1703 | 0.4443 |
> > > > >
> > > > > [1] Deep Neural Networks Constrained by Decision Rules, AAAI, 2019.

---

> > > ### Author Response · Authors · 2022-08-06
> > > **Additional Response to Reviewer pby1 [2/4]**
> > >
> > > We will also clarify that our primary contribution is not to advance the current field of neuro-symbolic reasoning by proposing a brand new logic rule framework significantly different from existing ones. Rather, we **leverage logic rule reasoning as a tool and apply it in explainable AI to significantly boost explanation quality**. Our main contribution consists of reflecting on the problem of explainable AI and considering the most suitable ways to integrate logic rule reasoning.  Nonetheless, we acknowledge there is still much to learn from other methods of neuro-symbolic reasoning. To clarify and avoid confusion, we will modify the second contribution (lines 84-86) of the Introduction as follows:
> > >
> > > *“We propose a logic rule reasoning framework that upgrades a given deep model into a self-explainable version by naturally integrating human priors, rule confidence modeling, and rule generation as an essential part for model prediction. Our method can achieve high human precision without depending on strong external knowledge such as instance-level guidance about rules, knowledge graphs, or a small number of rule candidates.”*
> > >
> > > >Scope of applicability and complexity of proposed solution based on neural network (why not decision trees?)
> > > > - **[Point 1]** Apart from the missing references, while reading the questions raised by reviewer oPY7, I realised that you manually engineer the candidate set used for the explanation (namely, you extract frequent words in text data, specific attribute values or quantized levels for tabular data). Indeed the neural network is only selecting elements from the candidate set and not really performing feature learning.
> > > > - Regarding point 1. The manual design of the candidate set is indeed a limiting factor. While I agree that the tasks you provided are suited for such assumption, is it still valid for other tasks involving for example image data? This limits the scope and applicability of the proposed strategy.
> > >
> > > We appreciate your insightful comments. Even if the common words are not chosen, our framework can still be used. In this case, we select the frequent words to incorporate a human prior (Sec. 2.3) because most of the low frequency words are not useful (the long tail effect) and using all of them results in a longer training time without performance improvement. This human prior is intuitive and easy for non-experts to understand. Moreover, the result is not sensitive to this prior, as demonstrated in our experiments with varying numbers of atoms (Supplement C.4.4 and Response to Reviewer zwRN [3/4]).
> > >
> > > **Our framework is applicable to various data types**, including NLP, tabular data, and graphs, all of which have distinct data units that can serve as atoms (e.g., words, features, nodes, and edges). **For image data, we can perform atom or feature learning by incorporating existing methods**. For example, in SENN [1], atoms represent a cluster of images or superpixels, and we can extract atoms with CAV (Concept Activation Vector) from [2], a feature that indicates the concept of humans (e.g., striped, red). It should be noted that such end-to-end feature learning is possible in our framework if the number of candidate atoms is small (e.g., around 100 object classes or concepts as in [3]). In this case, we could replace the training of the neural consequent estimator with real-time computation of rule confidence and learn the atoms (features) while also generating explanations. In the future, we will examine this intriguing direction.
> > >
> > > [1] Towards robust interpretability with self-explaining neural networks. NeurIPS 2018
> > >
> > > [2] Interpretability Beyond Feature Attribution: Quantitative Testing with Concept Activation Vectors (TCAV) ICML 2018
> > >
> > > [3] A Probabilistic Graphical Model Based on Neural-symbolic Reasoning for Visual Relationship Detection, CVPR, 2022

---

> > > ### Author Response · Authors · 2022-08-06
> > > **Additional Response to Reviewer pby1 [3/4]**
> > >
> > > >Technical Detail
> > > > - I would recommend to add more details in the main paper/supplementary about the losses and the hyperparameters used in all your experiments. Simply, providing a high-level description of losses and architectures is not enough to provide a complete description of your solution. However, I do appreciate that you are sharing the code.
> > > > - **[Point 2]** Furthermore, I don’t agree about the easiness of the proposed solution (transformer + autoregressive + transforme). Indeed, this is a rather complicated architecture, which brings in additional computational and optimisation issues.
> > >
> > > We thank the reviewer for this suggestion. We currently summarize the settings for important hyperparameters in supplement C.2 (lines 555-570), and provide a comprehensive list of all hyperparameters in the shared code file “utils.py.” We will extend the supplement to include a more detailed description of the hyperparameters and losses, which includes all of the hyperparameter settings, how we choose them, and to what extent they are sensitive.
> > >
> > > We would like to clarify that, with the exception of batch sizes and learning rate, most of the hyperparameters are set to the default values.  These hyperparameters are identical across all datasets. This, we believe, indicates that our model is relatively simple to optimize.
> > >
> > > *“The backbone models for textual data (BERT, RoBERTa) adhere to the original setting, and the model for tabular data (DNN) consists of a network with three fully-connected layers with a ReLU activation layer (FC-ReLU-FC-ReLU-FC) with 512 hidden dimensions. For deep antecedent generation, neural consequent estimation, and other baseline models, we set the hidden dimension $|h|$ as the default BERT and RoBERTa embedding size (i.e., 768) for textual data and 512 for tabular data. Cross-entropy loss is used to optimize the probability predicted by the consequent estimator for the antecedents extracted by the antecedent generator during the training of our model. For RCN, we extract a predefined rule set by following the original work. In particular, the predefined rules are decision paths in random forests with 100 estimators and a maximum depth of 4. After excluding stopwords, we fix the atoms in textual data to only come from the top-5000 frequent words. Tabular data uses both categorical and numerical features for atoms, while the threshold for numerical features is set to the 25th, 50th, and 75th percentiles of the data. The length of rule L (i.e., the number of atoms from recursive deep antecedent generation) is set to 4. The minimum document frequency is set to 200, and the number of rules for pretraining the neural consequent estimator is set to 10,000.*
> > >
> > > *We introduce the hyper-parameters of our model and baselines. Note that the same hyper-parameters are used for training baselines, the neural consequent estimator, and the deep antecedent generator for all datasets. The base backbone network and self-explainable models are trained 10 epochs. The batch size is set to 16, the largest size that can be trained on our GPU. For optimization, we employ Adam optimizer with a learning rate of 1e\-5, and ExponentialLR scheduler with  $\gamma$ 0.95. The learning rate with the best performance is selected after experiments on 5e\-5, 4e\-5, 3e\-5, 2e\-5, and 1e\-5. For SENN, a set of token embeddings from the pretrained language model (i.e., BERT and RoBERTa) are utilized as inputs and are considered to be interpretable basic concepts for textual data experiments. In the case of tabular data, raw input features are used. We follow the original work's implementation, and hyper-parameter settings for training, such as optimizer or scheduler, are all set to be identical to those of other baselines for a fair comparison.”*

---

> > > ### Author Response · Authors · 2022-08-06
> > > **Additional Response to Reviewer pby1 [4/4]**
> > >
> > > > Comparison to Decision Tree
> > > > - Regarding point 2. If your consider a boolean feature space over the manually defined candidate set (where the values refer to the presence or absence of elements in the data from the candidate set), then you can use a decision tree to select the most relevant elements for prediction. Importantly, it is possible to extract Boolean decision rules from the decision tree. Therefore, why should one prefer your solution over a decision tree (which is much simpler computationally speaking and also interpretable by nature)?
> > > Note also that this simple solution solves the issue about the self-consistency of explanations among different input samples, as raised by reviewer YUk5.
> > >
> > > Thank you for this comment. As shown in the table below, fully transparent models such as decision trees and random forests cannot achieve comparable prediction performance to deep models. Another option is to extract candidate rules with fully transparent models and use deep learning to compute the rule weight, which is **our baseline RCN [1]**. This method extracts rules using random forests and then computes the attention (rule weights) for all the rules using deep learning to predict each instance and generate explanations. **The prediction performance and explanation quality of this baseline is limited by the expressivity of traditional fully transparent models** (e.g., random forests, decision trees, or linear regression). While fully transparent models produce atoms or rules that meet the global data distribution, they frequently fail to adequately represent a single sample, especially on large datasets with many features. This can be seen from the fact that RCN is rarely chosen as the best explanation, particularly for Yelp text data (Table 3(c-d)). The finding suggests that directly using atoms or rules extracted with random forests introduces biases into deep models.
> > >
> > > Our method is **self-consistent** in the sense that 1) we ensure that **the prediction result for the same explanation is always the same**, e.g., the explanation “awesome” always leads to positive sentiment for different instances; and 2) our method **will not provide different explanations for similar inputs unless there is a solid reason** (e.g., the most confident rules happen to be different). (Please also see our response to Reviewer YUk5 [2/6] for the relevant theoretical analysis.) For example, if input-1 is changed to input-2 by substituting “very disappointing” with “disappointing,” then the best explanation changes from “very disappointing” in input-1 to “awful” in input-2, despite the fact that the two inputs are similar. Rather than forcing the explanations to be similar, our model chooses different explanations to achieve the best result. This process is analogous to humans’ “paying attention”  in the conscious Turing machine framework [2].
> > >
> > > [1] Deep neural networks constrained by decision rules. AAAI 2019
> > >
> > > [2] A theory of consciousness from a theoretical computer science perspective: Insights from the Conscious Turing Machine, PNAS, 2022
> > >
> > > **Table 1 Performance comparison between fully transparent model (random forest) and our model (RoBERTa and DNN are used for the backbone network).**
> > > |  | Yelp |  | Clickbait |  | Adult |  |
> > > |---|---|---|---|---|---|---|
> > > |  | F1 | AUC | F1 | AUC | F1 | AUC |
> > > | Random Forest | 73.03 | 80.40 | 44.29 | 60.25 | 65.602 | 66.146 |
> > > | Ours | 97.13 | 97.78 | 74.20 | 64.14 | 77.37 | 70.36 |

---

> ### Author Response · Authors · 2022-08-02
> **Response to Reviewer pby1 [2/3]**
>
> > Limitations of the chosen class of logic
> > - **[Weakness 2]** The work focuses on learning logical rules in the form of definite clause logic (a subclass of propositional logic), thus inheriting its limitations in terms of expressivity and also explainability. A discussion on these limitations is required.
> > - **[Limitation 1]** Are there any limitations on the chosen class of logic? How would this influence human precision?
>
> We thank the reviewer for this insightful comment. It is true that this work focuses only on the most basic class of logic, and that its expressivity and explainability could be further enhanced by considering more advanced classes of logic. In this paper, **we choose this class of logic because of three reasons**.
> - First, the simple definite clause logic has already achieved an impressive increase in expressivity and explainability compared with the currently widely-used explanations (e.g., [1], [2], [3]), which show the important words or features used for prediction. The definite clause logic is a natural extension to current explanations: it also considers words or features like “awesome” and “tasty” as the basic units, but can model a variety of feature interactions by using diverse logical connectives such as AND, OR, NOT, BEFORE. Its similarity with current explanations allows us to conduct a fair comparison with existing methods, and prove that such simple logic already achieves a much better human precision.
> - Second, as the first step, we hope that our method is easy to use and understand for researchers in explainable AI, who are not experts in logic programming. In this way, they may be more willing to follow this line of work. From this aspect, definite clause logic is perfect as it requires only the most basic and familiar terminologies (antecedents, consequence, atom, and logical connectives) and widely-used deep learning modules.
> - Third, we wish to increase the impact of our work by leaving large space for improvement and future research. Your suggestion on discussing the limitations is great for achieving this goal.
>
> We observe **two major limitations of using definite clause logic** as explanations.
> - The first limitation is the **coverage of the rules**. Since predicates are not considered, we cannot explicitly model higher level properties of atoms (e.g., learn that “awesome” is a “positive sentiment word” and make a rule based on “positive sentiment word”). Because our rules are constructed with bottom-level atoms (e.g., specific words), they usually have a low coverage (e.g., rule *“awesome AND tasty” => positive sentiment* only covers 0.37% of the input instances). Extending to first-order logic may naturally solve this problem.
> - The second limitation is the **level of insight** that we can obtain. Currently, we can only find rules constructed with bottom-level atoms instead of summarizing important high-level patterns. If upgraded to first-order logic, we may directly find high-level patterns such as *“a negation word AND a positive sentiment word” => negative sentiment*, instead of listing many specific rules such as *“not great” => negative sentiment* and *“no good” => negative sentiment*. This could save human cognitive budget and greatly improve the reasoning capability of deep models. Moreover, we may automatically compose high-level concepts such as “strong positive phrase” and build rules with them. The concept “strong positive phrase” may be composed by detecting two consecutive positive sentiment words (“amazingly comfortable” and “perfectly enjoyable”) with predicate invention in [4].
>
> While extending the method to other classes of logic is promising, there are still many **challenges** to solve before achieving this goal. For example, we need to study how to embed predicates in neural networks, how to handle the even larger solution space that contains a huge number of possible rules and clauses, and how to design the set of predicates without missing important reasoning logic or depending on biased heuristics. If such challenges can be solved, we believe that the **human precision** of logic rule reasoning will be further improved.
>
> We will discuss this and other limitations of our work in the revised version of the paper.
>
> [1] Towards robust interpretability with self-explaining neural networks. NeurIPS 2018
>
> [2] “Why Should I Trust You?” explaining the predictions of any classifier. KDD 2016
>
> [3] Anchors: High-precision model-agnostic explanations. AAAI 2018
>
> [4] Learning Explanatory Rule From Noisy Data. JAIR 2018

---

> ### Author Response · Authors · 2022-08-02
> **Response to Reviewer pby1 [3/3]**
>
> > **[Question 2]** What are the soft constraints and loss used in the experiments?
>
> Our work does not employ any soft constraints (i.e., soft prior) or loss (loss $L_b$ in Sec. 2.3) in the final implementation, as good explainability could be achieved without them. For classification, we employed the commonly-used cross-entropy loss. We will revise the paper to make it more clear.
>
> > **[Minor]** MINOR MISTAKE. Please replace $p(\alpha|b)$ with $p(b|\alpha)$  in Eq. (1), line 115, line 131, multiple occurrences in line 132, line 133, line 142 and Eq. (6). Also, modify the generative process in Figure 1 (c) to take into account the correct dependence between $\alpha$ and $b$.
>
> We thank the reviewer for these constructive suggestions. We will carefully incorporate these modifications into the revised paper.

---

### Official Review · Reviewer_oPY7 · 2022-07-11

**Rating:** 7
**Confidence:** 3
**Soundness:** 4 excellent
**Presentation:** 4 excellent
**Contribution:** 3 good

**Summary:**

This paper addresses the task of explaining deep models with logic form and proposes an explanation model applicable to text and tabular data, which generates latent antecedent and then makes a prediction based on the generated antecedent.
The experiments on review text and tabular datasets show that the proposed model outperforms the comparing methods in terms of classification performance and explainability.

**Questions:**

- How do you make the candidates of atoms? Focusing on the word 'awesome', many candidates of atoms associated with 'awesome' can be made, such as 'awesome$\geq 1$', 'awesome$\geq 2$', and so on.
- Is there a possibility that atoms associated with the same word, such as 'awesome$\geq 1$' and 'awesome$\geq 2$' are generated for the explanation in an instance?

**Strengths And Weaknesses:**

## Strengths
- The proposed model looks well designed, and the construction of each component of the proposed model, such as human prior, consequent estimation, and deep antecedent generation are reasonable.
-  As shown in Table 2, regardless of making the unexplainable models, such as BERT explainable, the proposed model can remain high classification performance.
- The paper is well written, and the performances of the proposed model in terms of classification performance and explainability are well analyzed with both numerical experiments and user study.

## Weaknesses
I don’t have any weaknesses to point out.

---

> ### Author Response · Authors · 2022-08-02
> **Title: Response to Reviewer oPY7**
>
> > **[Question 1]** How do you make the candidates of atoms? Focusing on the word 'awesome', many candidates of atoms associated with 'awesome' can be made, such as 'awesome ≥ 1', 'awesome ≥ 2', and so on.
>
> Thanks for the question. We make the atoms so that they have a consistent form with most of the baselines, in order to ensure a fair comparison. Specifically, in current implementation, we only consider the atoms like “awesome≥ 1” for textual datasets. This enables the comparison with the explainable models that highlights the words based on their importance weight. For tabular datasets, we choose different strategies based on feature types. For categorical features, whether the instance belongs to a certain category or not becomes an atom. For example, in the Adult dataset, “marital-status == Married” would indicate the person in the given instance is married. For numerical features, we calculate 25, 50, 75 percentile of overall feature distribution for the threshold. We use whether the feature of a given sample is larger or smaller than the threshold as atoms. For example, the feature “age” of the Adult dataset has thresholds 28, 37, 48. Thus, we made atoms  “age ≥ 28”,  “age < 28”,  “age ≥ 37”,  “age < 37”,  “age ≥ 48”,  “age < 48”. This is consistent with the atoms in our baseline RCN [1], which are created by random forests. We will explain this more clearly in the paper.
>
> [1] Deep neural networks constrained by decision rules. AAAI 2019
>
> > **[Question 2]** Is there a possibility that atoms associated with the same word, such as 'awesome ≥ 1' and 'awesome ≥ 2' are generated for the explanation in an instance?
>
> Thanks for the question. There is a possibility that different atoms associated with the same word or feature appear in the same explanation, when there exist atoms that overlap with each other. This is the situation for our tabular datasets (e.g., “age ≥ 37” and “age ≥ 48” for the feature “age”). In such a situation, we remove the redundant atoms after the explanation has been generated (e.g., removing “age ≥ 37” if “age ≥ 48” is generated). Note that the generated atoms will not be conflicted with each other. For example, “age ≥ 48” and “age<37” will not be generated simultaneously in one explanation, because the condition for generation is that the corresponding instance satisfies both atoms. This is enforced by the local constraint introduced in Sec. 2.5. We find that such a post-processing step of removing redundant atoms is easy to implement and has reasonably good explainability and prediction performance. It is also possible to eliminate redundant atoms during explanation generation. One possible way is to create the atoms so that they do not overlap (e.g., creating “age ≥ 48”, “48 > age ≥ 37”, “37 > age ≥ 28”, “28 > age” for feature “age”). However, this may make it impossible to flexibly combine different thresholds (e.g., generating “48 > age ≥ 28”). Another way is to apply a mask to the model so that it assigns zero probability to an already chosen feature or a redundant atom. This can be implemented by carefully setting the local constraint in Sec. 2.5. We will discuss more about this in the supplement.

---

### Official Review · Reviewer_YUk5 · 2022-07-11

**Rating:** 6
**Confidence:** 4
**Soundness:** 3 good
**Presentation:** 2 fair
**Contribution:** 3 good

**Summary:**

The paper describes a self-explainable model in which explanations consist of logical rules consistent with the data and decisions and decisions are consistent with these explanations (at least locally).  Both elements are implemented using neural networks.  The model also supports prior knowledge on admissible rules.  Results on a selection of textual and tabular data sets illustrate the performance and robustness of the proposed approach compared to black-box models, post-hoc explainers, and self-explainable alternatives.

**Questions:**

- See Q1 and Q2 above.

- Are the results reported for the human study statistically significant?

- l 279: "The participants are allowed to choose multiple best explanations only if the chosen ones are the same"  Why?  Can't two explanations be equally good (or bad)?

**Limitations:**

Partially, in the conclusion.  Other limitations include lack of generalization of produced explanations across inputs and lack of self-consistency of explanations at different points.  I asked the authors to comment on these in my questions above.

**Strengths And Weaknesses:**

**Originality**: The proposed approach combines familiar elements into a novel architecture.

**Clarity**: English is good, but the text - and especially the description of the method - is partially opaque.  Notation is not entirely clear.  This makes it hard to assess the properties of the proposed model.

In particular, Eq. 1 is confusing as it looks like the explanations are being marginalized over, which would mean that any one prediction could be the result of potentially exponentially many explanations (all those appearing in the sum and having probability > 0).  This would conflict with the goal of building an interpretable model.  To the best of my understanding, however, this is not the case, as both p(y|alpha) and p(alpha|y) are essentially discriminative, so no real marginalization takes place at inference time.  This is not deal-breaking, but it is an issue.

**Quality**: At a high-level, the proposed model is reasonable.  One issue I have with it is that it looks very complex - much more so than SENNs or concept bottleneck models.

Q1. Another element of confusion is the relationship between rules for different inputs:  is it possible for close inputs to have arbitrarily different explanations?  In other words, what is the ``radius of validity'' of the local explanations produced by the model?

Q2. More generally, is it possible that the rules predicted by the model are inconsistent with each other, i.e., that the rule predicted for an input x and that *should* apply to input x' is inconsistent with the rule predicted for input x'?

**UPDATE**: Q1 AND Q2 HAVE BEEN ANSWERED BY THE AUTHORS IN THEIR REBUTTAL.

Lack of explanation generalization across points would complicate global interpretability (and woese self consistency) of the model.

The numerical experiments look okay, with a reasonable selection of data sets and a number of competitors.  One issue is that the human evaluation is *very* small scale - it only involves three human subjects.  Platforms like Prolific however make it quite easy to setup larger scale experiments at a very reasonable cost, so the choice of restricting the experiment to only three people is puzzling.  I am not sure that the numbers reported have statistical significance.  See questions below.

**UPDATE**: THE AUTHORS HAVE REPORTED MORE EXTENSIVE EXPERIMENTS IN THEIR REBUTTAL.

As for related work, the authors write: "Collecting ground-truth labels of human decision processes are quite expensive, if not impossible."  But there exist a number of ML approaches that do exactly that - using post-hoc and self-explainable models.  See for instance:

  Kulesza et al. "Principles of Explanatory Debugging to Personalize Interactive Machine Learning" 2015.

  Schramowski et al. "Making deep neural networks right for the right scientific reasons by interacting with their explanations" 2020.

  Lertvittayakumjorn et al. "FIND: Human-in-the-loop Debugging Deep Text Classifiers" 2020.

  Ciravegna et al. "Human-driven FOL explanations of deep learning" 2020.

  Stammer et al. "Right for the Right Concept: Revising Neuro-Symbolic Concepts by Interacting with their Explanations" 2021.

  Bontempelli et al. "Toward a Unified Framework for Debugging Gray-box Models" 2022.

just to mention a few.  I would expect there to be also works on acquiring rules from human annotators in the context of logical and neuro-symbolic models, but I cannot provide pointers.  I would have expected better coverage of works on debugging neural networks and self-explainable models.


**Significance**: The model is of interest to research in XAI and especially for self-explainable models.

Other comments:

- l 121: "The deep model f is used to encode the sample x"  Wasn't f the complete model?  Also, at this point it is not clear what
  a "recursive generation layer" is.  I'd suggest to move lines 121-123 to a later section.

- l 125: "Our framework for deep logic rule reasoning bears an interesting resemblance to the human mind..."  All concept-based models and many neuro-symbolic models do.  But it's only a vague analogy and I'm not sure what it adds.  Moreover, notice that post-hoc models bear a striking resemblance to models of the mind in which decisions are taken subconsciously and then interpreted after the fact by an argumentative interpreter (see books by Michael Gazzaniga).  There is no conclusive evidence of how our minds really work in this regard.  Also, is really the subconscious mind "often thought to be more powerful"?

- l 183: square brackets, missing comma.

- l 191: organizes

- Eq 3.  The equality symbols should be replaced by \approx, especially the second one (as p is replaced by \hat{p}).

---

> ### Author Response · Authors · 2022-08-02
> **Response to Reviewer YUk5 [1/6]**
>
> > **[Clarity Weakness]** English is good, but the text - and especially the description of the method - is partially opaque. Notation is not entirely clear. This makes it hard to assess the properties of the proposed model.
> In particular, Eq. 1 is confusing as it looks like the explanations are being marginalized over, which would mean that any one prediction could be the result of potentially exponentially many explanations (all those appearing in the sum and having probability > 0). This would conflict with the goal of building an interpretable model. To the best of my understanding, however, this is not the case, as both p(y|alpha) and p(alpha|y) are essentially discriminative, so no real marginalization takes place at inference time. This is not deal-breaking, but it is an issue.
>
> We thank the reviewer for making this important comment. We would like to clarify that **Eq. (1) and the statement that “no real marginalization takes place" are both correct, as Eq. (1) is approximated using sampling:**
>
> $p(y|x,b) \propto \sum_{\alpha}{p(b|\alpha)p(y|\alpha)p(\alpha|x)} = E_{\alpha \sim p(\alpha|x)} {p(b|\alpha)p(y|\alpha)} \approx {1 \over S} \sum_{s \in [1,S], \alpha^{(s)} \sim p(\alpha|x)} {p(b|\alpha^{(s)})p(y|\alpha^{(s)})}$
>
> where the first two terms are Eq. (1), the third term reformulates the second term as the expected value, and the fourth term is sampling-based approximation. Here, $S$ is the total number of samples, and $\alpha^{(s)}$ is a sample from the distribution $p(\alpha|x)$. Accordingly, $p(y|x,b)$ can be obtained by sampling a few explanations, without real marginalization.
>
> Given a small sample size ($S=1$ in this paper), **optimizing the fourth term naturally ensures the model interpretability**. Specifically, to maximize the fourth term $ {1 \over S} \sum_{s \in [1,S], \alpha^{(s)} \sim p(\alpha|x)} {p(b|\alpha^{(s)})p(y|\alpha^{(s)})}$ when $S=1$, the explanation generator $p(\alpha|x)$ must find a single sample $\alpha^{(s)}$ that yields the largest $p(b|\alpha^{(s)})p(y|\alpha^{(s)})$, i.e., find $\alpha^{(s)}$ that maximizes prediction accuracy $p(y|\alpha^{(s)})$ and fitness to human prior $p(b|\alpha^{(s)})$. Furthermore, the generator must assign a very large probability to the best $\alpha^{(s)}$, otherwise, other samples with a lower $p(b|\alpha^{(s)})p(y|\alpha^{(s)})$ may be generated, thereby decreasing $p(y|x,b)$. This ensures the sparsity of $p(\alpha|x)$ and the model interpretability. If there are multiple best explanations that result in the same $p(b|\alpha^{(s)})p(y|\alpha^{(s)})$, the explanation generator may find all of them.
> Sampling and interpretability are described in Sec. 2.6 and Eq. (6) of the paper (lines 208-218). We will revise the paper in accordance with the reviewer’s suggestion and introduce this part earlier in the paper (e.g., along with Eq (1)).
>
> > **[Quality Weakness 0]** At a high-level, the proposed model is reasonable. One issue I have with it is that it looks very complex - much more so than SENNs or concept bottleneck models.
>
> Thank the reviewer for this valuable feedback. We recognize the conceptual complexity of logic rules and their integration into a Bayesian framework. We will do our best to improve readability by refining the discussion of the model framework based on the suggestions, using more examples for illustration, and replacing symbols with natural language descriptions whenever possible. Other suggestions on improving the readability will be highly appreciated.

---

> > ### Comment · Reviewer_YUk5 · 2022-08-04
> > **Response to author's response 1/6**
> >
> > > We would like to clarify that Eq. (1) and the statement that “no real marginalization takes place" are both correct, as Eq. (1) is approximated using sampling.
> >
> > > Given a small sample size ($S = 1$ in this paper), optimizing the fourth term naturally ensures the model interpretability.
> >
> > This is very clear, thank you.  My point was that if $S > 1$ then it is hard to claim that the inference process is interpretable: (1) if $S$ is large (say, larger than $25$ or so), then probably no human can really understand what is going on, and (2) if $S > 1$, then a particular prediction may result from marginalizing over mutually inconsistent explanations, again complicating interpretability.  So in practice $S$ will be restricted to a be very small number.  Now, I am okay with using $S = 1$ (as you do in the paper), but since $S$ is so small (and will likely always be very small, for the aforementioned reasons) I find Eq. 1 not very representative of what the actual inference process looks like.
> >
> > > One issue I have with it is that it looks very complex.
> >
> > I apologize, this was not quite clear enough.  Let me start by saying that model complexity is *not* a big issue for me.  Regardless, what I was implying is that I would expect your model to be harder to train (in terms of, e.g., stability) than more straightforward models like SENNs or CBMs.  I would appreciate if you could comment on how easy or hard it was to get your model to work as intended in your experiments.  I mean - is it really plug-and-play or did you have to play around with the choice of architecture, hyperparameters, etc.?

---

> > > ### Author Response · Authors · 2022-08-06
> > > **Additional Response to Reviewer YUk5**
> > >
> > > > Effect of The Number of Samples for Explanation ($S$)
> > > > - **We would like to clarify that Eq. (1) and the statement that “no real marginalization takes place" are both correct, as Eq. (1) is approximated using sampling. Given a small sample size (S=1 in this paper), optimizing the fourth term naturally ensures the model interpretability.**
> > > > - This is very clear, thank you. My point was that if S>1 then it is hard to claim that the inference process is interpretable: (1) if S is large (say, larger than 25 or so), then probably no human can really understand what is going on, and (2) if S>1, then a particular prediction may result from marginalizing over mutually inconsistent explanations, again complicating interpretability. So in practice S will be restricted to a be very small number. Now, I am okay with using S=1 (as you do in the paper), but since S is so small (and will likely always be very small, for the aforementioned reasons) I find Eq. 1 not very representative of what the actual inference process looks like.
> > >
> > > Thank you very much for your assistance in making the paper more readable. Eq. (1) will be revised to explicitly include the constraint for ensuring that $p(\alpha|x)$ is sparse (so that it can be approximated well with a limited number of samples). We will also discuss how approximating Eq. (1) with sampling achieves this goal automatically, as well as how different choices of $S$ affect interpretability.
> > >
> > > >One issue I have with it is that it looks very complex.
> > > > - I apologize, this was not quite clear enough. Let me start by saying that model complexity is not a big issue for me. Regardless, what I was implying is that I would expect your model to be harder to train (in terms of, e.g., stability) than more straightforward models like SENNs or CBMs. I would appreciate if you could comment on how easy or hard it was to get your model to work as intended in your experiments. I mean - is it really plug-and-play or did you have to play around with the choice of architecture, hyperparameters, etc.?
> > >
> > > We appreciate this practical feedback. It is relatively simple to obtain a good result with our method. Most hyperparameters are set to their default values. We adjusted the learning rate and batch size to the largest that works in our GPU, for the Yelp dataset. We applied the same hyperparameter settings to all datasets (5 datasets in total).
> > >
> > > We also confirm that our method is not affected by hyperparameters or model architectures. We tested GRU and Transformer as antecedent generators and found their performance to be similar. Our model is unaffected by other hyperparameters such as antecedent length, atom candidate count, and sample count in consequent estimator pretraining. Analysis of these parameters is provided in the supplement. In response to your feedback, we will include more information about hyperparameters in the paper.
> > >
> > > Based on these observations, we think there is a good chance that for a new dataset, users can directly plug in our method with the current hyperparameter settings and obtain an acceptable result.

---

> ### Author Response · Authors · 2022-08-02
> **Response to Reviewer YUk5 [2/6]**
>
> > Generality and self-consistency of explanations
> > - **[Quality Weakness 1]**  Another element of confusion is the relationship between rules for different inputs: is it possible for close inputs to have arbitrarily different explanations? In other words, what is the ``radius of validity'' of the local explanations produced by the model?
> > - **[Quality Weakness 2]** More generally, is it possible that the rules predicted by the model are inconsistent with each other, i.e., that the rule predicted for an input x and that should apply to input x' is inconsistent with the rule predicted for input x'?
> > - **[Quality Weakness 3]** Lack of explanation generalization across points would complicate global interpretability (and woese self consistency) of the model.
> > - **[Limitations]** Other limitations include lack of generalization of produced explanations across inputs and lack of self-consistency of explanations at different points. I asked the authors to comment on these in my questions above.
>
> We appreciate the reviewer’s feedback. Our method is **self-consistent** in the sense that 1) we ensure the prediction result for the same explanation is always the same, e.g., the explanation “awesome” always leads to positive sentiment for different instances; and 2) our method will not provide different explanations for similar inputs unless the best rules for the two inputs indeed change, because our rule generator is trained to optimize two globally consistent rewards.  For example, if input-1 is changed to input-2 by substituting “very disappointing” with “disappointing,” then the best explanation indeed changes from “very disappointing” in input-1 to “awful” in input-2, even though the two inputs may be considered similar. In this way, we will pick different explanations to achieve the best result, rather than forcing the explanations to be similar. This process is analogous to humans’ “paying attention” in the conscious turing machine framework [1]. We provide detailed theorectical analysis as follows.
>
> For linear-regression-based models like SENN, the explanations for similar inputs may be entirely different without specific constraints like the robustness loss, because the main optimization goal for SENN is the **local prediction accuracy**. Without the robustness loss, the model may find a correct prediction locally for a single instance, but being “surely no more interpretable than any deep neural network” (quoted from the SENN paper [2]).
>
> However, this is not the case for the logic rule reasoning framework, because the rule generator is trained to optimize **two globally consistent rewards** (Eq. (6) and Sec. 2.6): **human’s prior belief** about which explanation types are good and **the explanation confidence** that is measured by the global prediction accuracy over the entire training corpus given the explanation (rule). Thus, explanations for similar inputs may be different only when:
>
> 1) the optimal (most confident and human-preferred) rules for the inputs are different;
>
> 2) there are multiple explanations that achieve the exact same reward; or
>
> 3) the model has not been trained sufficiently to achieve the optimal result.
>
> In situation 1), our method removes the heuristic constraint regarding the similarity of explanations, allowing us to identify the optimal (e.g., most confident) explanations for the two inputs. For example, if input-1 is changed to input-2 by substituting “very disappointing” with “disappointing,” then the best explanation may change from “very disappointing” in input-1 to “awful” in input-2. Even if the two inputs are similar, their optimal explanations may differ in this instance. This is plausible as such a change increases the explanation’s confidence. In other words, the **radius of validity** of an explanation corresponds to inputs that have similar optimal (most confident and human preferred) rules. For example, explanation *“very disappointing”=>negative sentiment* can **generalize** to all instances that satisfy the rule and at the same time do not satisfy a more confident rule. When we want to force the explanations of two inputs to be similar, we can also incorporate a constraint that mimics the robustness loss in SENN into the soft human prior by defining $L_b$ (Sec. 2.3). We will discuss this in the paper as an exciting future direction. Situation 2) rarely occurs, as our explanation confidence reward $p(y|\alpha)$ is a real number, not a discrete value. In rare cases where this occurs, it is possible to remedy the situation by using the soft human prior. Situation 3) can be avoided by checking the training loss, the classification accuracy (Sec. 3.2), and the explainability (Sec. 3.3).
>
> [1] A theory of consciousness from a theoretical computer science perspective: Insights from the Conscious Turing Machine, PNAS, 2022
>
> [2] Towards robust interpretability with self-explaining neural networks. NeurIPS 2018
>
> *(not finished, see the next message: response [3/6])*

---

> > ### Comment · Reviewer_YUk5 · 2022-08-07
> > **Reply to 2/6**
> >
> > Thank you for your replies and apologies for reacting so late, I have been traveling & had to deal with some hardware issues.
> >
> > > this is not the case for the logic rule reasoning framework, because the rule generator is trained to optimize two globally consistent rewards (Eq. (6) and Sec. 2.6): human’s prior belief about which explanation types are good and the explanation confidence that is measured by the global prediction accuracy over the entire training corpus given the explanation (rule). Thus, explanations for similar inputs may be different only when:
> >
> > Let's see if I got this right:
> >
> >  1. The human prior affects what antecedents $\alpha$ are to be preferred for every input $x$.
> >
> >  2. The global prediction accuracy makes class predictions consistent with each other once the antecedent $\alpha$ has been generated.
> >
> > My view is that these two mechanism have more to do with the choice of $\alpha$ independently from the input $x$ (the prior) and with the choice of $y$ given $\alpha$, but have little to do with the choice of $\alpha$ given $x$.  So it may still be the case that similar $x$'s are associated to different $\alpha$'s.  Is this correct?
> >
> > > Even if the two inputs are similar, their optimal explanations may differ in this instance. This is plausible as such a change increases the explanation’s confidence.
> >
> > Sure.  For instance, in decision trees very close inputs may be covered by different leaves with very different paths.  Also, out of all concept-based models, only SENNs really care about this.  So this is fine by me.
> >
> > The reason why I brought these issues up is they are not specifically mentioned in the paper - so readers and end-users may end up getting the wrong impression about what the model is designed to do.  I think that having a short paragraph in which you very clearly explain that similar points may *not* get similar explanations, that no mechanism is specifically put in place to prevent this, and that this *may* be okay depending on the application and user expectation, would be enough for me.  All I am asking is to be explicit about this in the text.
> >
> > Another option is that you prove me wrong (which could very well be) by reporting statistics concerning how unlikely similar instances are associated different explanations in your experiments.
> >
> > I'd be happy to increase my score once either of these options is carried out.

---

> > > ### Author Response · Authors · 2022-08-08
> > > **Additional Response to Reviewer YUk5 on Reply to (2/6)**
> > >
> > > >Clarification on mechanisms of our model
> > > > - Let's see if I got this right:
> > > > 1. The human prior affects what antecedents α are to be preferred for every input x
> > > > 2. The global prediction accuracy makes class predictions consistent with each other once the antecedent α has been generated.
> > > > - My view is that these two mechanism have more to do with the choice of α independently from the input x (the prior) and with the choice of y given α, but have little to do with the choice of α given x. So it may still be the case that similar x's are associated to different α's. Is this correct?
> > >
> > > Yes, your understanding about the two mechanisms is right. Sorry for not explaining our idea clearly in our previous response. We aimed to illustrate that if similar inputs are defined as *inputs whose best rules are similar* (the most confident and human preferred rules satisfied by the inputs are similar), then we can say that our method will generate similar explanations for similar inputs. This can be seen as a similarity computed after removing less-important features (e.g., less-confident atoms). However, this conclusion is made only according to idealized theoretically analysis, in which we assume that Eq. (1) is fully optimized and that there do not exist multiple rules with exactly the same confidence score. Moreover, we can not guarantee the stability of explanations given other definitions of similarity, e.g., *similarity between the raw features*.
> > >
> > > > Clarification on explanation consistency
> > > > - Sure. For instance, in decision trees very close inputs may be covered by different leaves with very different paths. Also, out of all concept-based models, only SENNs really care about this. So this is fine by me.
> > > > - The reason why I brought these issues up is they are not specifically mentioned in the paper - so readers and end-users may end up getting the wrong impression about what the model is designed to do. I think that having a short paragraph in which you very clearly explain that similar points may not get similar explanations, that no mechanism is specifically put in place to prevent this, and that this may be okay depending on the application and user expectation, would be enough for me. All I am asking is to be explicit about this in the text.
> > > > - Another option is that you prove me wrong (which could very well be) by reporting statistics concerning how unlikely similar instances are associated different explanations in your experiments.
> > >
> > > Thanks for your detailed and constructive feedback. We will explicitly discuss this in the paper based on your suggestion. Here are our current descriptions. Your further suggestions on revising this paragraph are always welcomed.
> > >
> > > *“A key desirable property for self-explaining models is stability [1], which requires that similar inputs result in similar explanations. Unlike SENN [1], which proposes a robustness loss to ensure that explanations are stable with respect to adversarial inputs, our method does not employ such a constraint and cannot guarantee the stability of explanations for inputs that have similar raw features. Instead, our stability is modeled in a selected feature space, in which only the features in the optimal rule are considered for an input. Specifically, given an input $x$ “The restaurant is awesome and the food is incredibly tasty,” its selected features are $x’$ “awesome, incredibly, tasty” that compose the most confident and human preferred rule. In our framework, any inputs with similar $x’$ will have similar explanations theorectically. In other words, any change in $x’$ (instead of $x$) will result in a change in explanations, while other changes will not. Note that this statement about stability is made only according to idealized theoretically analysis, in which we assume that Eq. (1) is fully optimized and there do not exist multiple rules with exactly the same confidence score. We are interested in evaluating the stability of our method in both the selected and raw feature space with comprehensive experiments, and will try integrating a similar constraint with that in SENN [1] into our soft human prior in the future.”*
> > >
> > > [1] Towards Robust Interpretability with Self-Explaining Neural Networks, NeurIPS, 2018

---

> ### Author Response · Authors · 2022-08-02
> **Response to Reviewer YUk5 [3/6]**
>
> In addition to theoretical analysis, we give examples of generated explanations for similar inputs to show how generalizable our explanation is (see the following tables). More cases will be given in the supplement, where the instances are randomly sampled from the test set and manually modified to create similar samples. We **highlight** the part of the sentence that has been changed manually to create a similar input.
>
> In addition, our case study (Sec. 3.3) also shows that our explanations provide **a good global interpretability** that is helpful for model debugging and refinement. More specifically, we show that studying the explanation clusters helps quickly identify a group of problematic input instances and potential reasons for the relatively low performance.
>
> | Sample | Model Explanation | Prediction |
> |---|---|---|
> | This place is awful. The pizza was cold, and the waiters were rude. | awful, cold, rude | Negative |
> | This place is awful. The __pasta__ was cold, and the __servers__ were rude. | awful, cold, rude | Negative |
> | This place is awful. The pizza was **undercooked**, and the waiters were **unfriendly**. | awful, undercooked, unfriendly | Negative |
>
> | Sample | Model Explanation | Prediction |
> |---|---|---|
> | I love here. It was an amazing experience to eat a cheesy macaroni. | love, amazing, cheesy | Positive |
> | I **recommend** here. It was **a happy** experience to eat a cheesy macaroni. | recommend, happy, cheesy | Positive |
> | I **hate** here. It was **a bad** experience to eat a cheesy macaroni. | hate, bad, experience | Negative |
>
> | Sample | Model Explanation | Prediction |
> |---|---|---|
> | I ordered three tacos and all 3 were downright lousy. Can't remember the last time I had food this bad. The shrimp taco was overbreaded and in a sickly sweet sauce, the shredded beef taco was very tiny and thankfully, I can't remember what the third taco tasted like. To the reviewer who posted that these tacos are top notch.... what are you smoking? I waited forever to get my food and saw numerous other people who came in after me get their food.  Waiter was MIA. Not coming back....ever. | lousy, bad, waited, forever | Negative |
> | I ordered three tacos and all 3 were downright lousy. Can't remember the last time I had food this bad. The shrimp taco was overbreaded and in a sickly sweet sauce, the shredded beef taco was very tiny and thankfully, I can't remember what the third taco tasted like. To the reviewer who posted that these tacos are top notch.... what are you smoking? I waited **a little** to get my food and saw numerous other people who came in after me get their food.  Waiter was MIA. Not coming back....ever. | lousy, bad, waited, not | Negative |
> | I ordered three **awful, terrible** tacos and all 3 were downright lousy. Can't remember the last time I had food this bad. The shrimp taco was overbreaded and in a sickly sweet sauce, the shredded beef taco was very tiny and thankfully, I can't remember what the third taco tasted like. To the reviewer who posted that these tacos are top notch.... what are you smoking? I waited forever to get my food and saw numerous other people who came in after me get their food.  Waiter was MIA. Not coming back....ever. | awful, terrible, waited, forever | Negative |
>
> | Sample | Model Explanation | Prediction |
> |---|---|---|
> | I had an amazing 4 course meal here with my family from philadlephia. my father runs a farmers market there and was very impressed with their use of seasonal and local foods. We had an amazing pork belly salad and I had duck wrapped in bacon and stuffed with pate, which sounds insanely heavy, but it was not; the portion was small enough not to be overwhelmed and it was not overly greasy at all. It was a fantastic meal. I think l'etoile is on par with top restaurants in bigger cities. | amazing, family, stuffed,  fantastic | Positive |
> | I had an **great** 4 course meal here with my family from philadlephia. my father runs a farmers market there and was very impressed with their use of seasonal and local foods. We had an amazing pork belly salad and I had duck wrapped in bacon and stuffed with pate, which sounds insanely heavy, but it was not; the portion was small enough not to be overwhelmed and it was not overly greasy at all. It was a fantastic meal. I think l'etoile is on par with top restaurants in bigger cities. | great,  family, amazing,  fantastic | Positive |
> | I had an **awful** 4 course meal here with my family from philadlephia. my father runs a farmers market there and was very **disappointed** with their use of seasonal and local foods. We had a **terrible** pork belly salad and I had duck wrapped in bacon and stuffed with pate, which sounds insanely heavy; **and it was right**;  the portion was **too small to be full** and it **was overly** greasy at all. It was a **bad** meal. I think l'etoile is on par with **bad** restaurants in bigger cities. | awful, disappointed, terrible, bad | Negative |

---

> ### Author Response · Authors · 2022-08-02
> **Response to Reviewer YUk5 [4/6]**
>
> > Questions about user study
> > - **[Quality Weakness 4]** The numerical experiments look okay, with a reasonable selection of data sets and a number of competitors. One issue is that the human evaluation is very small scale - it only involves three human subjects. Platforms like Prolific however make it quite easy to setup larger scale experiments at a very reasonable cost, so the choice of restricting the experiment to only three people is puzzling. I am not sure that the numbers reported have statistical significance.
> > - **[Question 2]** Are the results reported for the human study statistically significant?
>
> We appreciate the reviewer's insightful suggestion. User studies can be performed by 1) hiring a large number of labelers from platforms like Prolific and AMT or 2) hiring a limited number of experienced annotators from a labeling company. While platforms like Prolific make it easy to find many labelers, they are known to be **better suited for cognitively simple tasks and may suffer from errors** [1]. Our task is challenging for ordinary labelers, as we require them to carefully reason about which features of an adult are useful for predicting his or her income (the Adult dataset) and compare multiple similar explanations. Thus, we validated the model with more experienced annotators hired through a labeling company. To ensure the labelers have an adequate understanding of the task, we provided them with detailed guidelines and examined their initial labels with feedback when a misunderstanding is detected. **Such a close interaction would not be possible in crowdsourcing platforms**, which may lead to errors and unreliable results.
> [1] Data quality of platforms and panels for online behavioral research, Behavior Research Methods 2021
>
> With the high quality labels, we can get a good agreement and **statistically significant results in most cases** even when the number of participants is small. Tables 1 to 4 show the result of the user study with p-values. We marked one star (*) if the p-value is lower than 0.1, and two stars (**) if it is less than 0.05. The result confirms that we obtain a reasonably good p-value (<0.1) in all cases except for LIME on Yelp.
>
> **Table 1 Percentage of Good Explanations with Previous Three Pariticipants (Yelp)**
> |  | Lime | Anchor  | SENN | RCN | Ours |
> |---|---|---|---|---|---|
> | Average | 88.7 | 84.0 | 22.0 | 36.0 | 90.0 |
> | Agreement | 86.7 | 85.3 | 90.7 | 54.7 | 89.3 |
> | P-value (Compared with Ours) | 6.56 E-1 | 7.17 E-2 * | 2.27 E-24  ** | 2.14 E-37  ** | N/A |
>
> **Table 2 Percentage of Best Explanations with Previous Three Pariticipants (Yelp)**
> |  | Lime | Anchor  | SENN | RCN | Ours |
> |---|---|---|---|---|---|
> | Average | 30.0 | 26.7 | 3.3 | 2.0 | 40.0 |
> | Agreement | 84.0 | 84.0 | 98.7 | 100.0 | 86.7 |
> | P-value (Compared with Ours) | 1.32 E-1 | 4.51 E-2 ** | 2.57 E-15 ** | 1.11 E-13 ** | N/A |
>
> **Table 3 Percentage of Good Explanations with Previous Three Pariticipants (Adult)**
> |  | Lime | Anchor  | SENN | RCN | Ours |
> |---|---|---|---|---|---|
> | Average | 66.7 | 33.3 | 64.7 | 78.0 | 87.3 |
> | Agreement | 50.7 | 84.0 | 50.7 | 78.7 | 89.3 |
> | P-value (Compared with Ours) | 3.60 E-5 ** | 2.27 E-24 ** | 1.58 E-2 ** | 2.37 E-5 ** | N/A |
>
> **Table 4 Percentage of Best Explanations with Previous Three Pariticipants (Adult)**
> |  | Lime | Anchor  | SENN | RCN | Ours |
> |---|---|---|---|---|---|
> | Average | 0.7 | 20.0 | 8.0 | 9.3 | 62.0 |
> | Agreement | 98.7 | 92.0 | 96.0 | 85.3 | 76.0 |
> | P-value (Compared with Ours) | 1.79 E-31 ** | 1.21 E-17 ** | 2.00 E-9 ** | 3.69 E-19 ** | N/A |
>
> *(not finished, see the next message: response [5/6])*

---

> ### Author Response · Authors · 2022-08-02
> **Response to Reviewer YUk5 [5/6]**
>
> To further validate the explainability of our method, we **hire 6 more participants** in response to your suggestion. The new user study result with these 6 participants can be found in Tables 5~8, which shows statistically significant difference (p-value<0.05) in all cases.
>
> We will add the result and related discussion in the paper.
>
> **Table 5 Percentage of Good Explanations with New 6 Participants (Yelp)**
> |  | Lime | Anchor  | SENN | RCN | Ours |
> |---|---|---|---|---|---|
> | Average | 90.3 | 84.7 | 40.7 | 78.0 | 96.7 |
> | Agreement | 84.4 | 87.7 | 72.3 | 77.6 | 93.87 |
> | P-value (Compared with Ours) | 8.69 E-4 ** | 1.12 E-7 ** | 1.40 E-51 ** | 7.26 E-13 ** | N/A  |
>
> **Table 6 Percentage of Best Explanations with New 6 Participants (Yelp)**
> |  | Lime | Anchor  | SENN | RCN | Ours |
> |---|---|---|---|---|---|
> | Average | 36.3 | 13.7 | 2.0 | 2.0 | 50.0 |
> | Agreement | 67.6 | 83.6 | 96.3 | 96.3 | 64.8 |
> | P-value (Compared with Ours) | 8.87 E-3 ** | 5.63 E-18 ** | 6.84 E-40 ** | 6.84 E-40 ** | N/A |
>
> **Table 7 Percentage of Good Explanations with New 6 Participants (Adult)**
> |  | Lime | Anchor  | SENN | RCN | Ours |
> |---|---|---|---|---|---|
> | Average | 30.7 | 62.3 | 36.7 | 52.3 | 92.3 |
> | Agreement | 57.1 | 59.9 | 51.5 | 53.2 | 85.7 |
> | P-value (Compared with Ours) | 6.09 E-54 ** | 5.56 E-18 ** | 1.18 E-41 ** | 2.83 E-27 ** | N/A |
>
> **Table 8 Percentage of Best Explanations with New 6 Participants (Adult)**
> |  | Lime | Anchor  | SENN | RCN | Ours |
> |---|---|---|---|---|---|
> | Average | 1.7 | 10.7 | 10.3 | 10.7 | 66.7 |
> | Agreement | 96.7 | 82.9 | 83.3 | 82.9 | 58.4 |
> | P-value (Compared with Ours) | 1.72 E-64 ** | 1.23 E-35 ** | 2.30 E-36 ** | 1.23 E-35 ** | N/A |

---

> > ### Comment · Reviewer_YUk5 · 2022-08-07
> > **Reply to 4/6 and 5/6**
> >
> > Thank you for setting up these extra experiments and for reporting the statistical significance over this larger sample size.  The new updated tables to still be very much in favour of the proposed approach.  This makes me much more comfortable in supporting acceptance.
> >
> > Assuming that the points raised in 2/6 will be addressed (they are an easy fix), I will proceed to increase the score of the paper.

---

> ### Author Response · Authors · 2022-08-02
> **Response to Reviewer YUk5 [6/6]**
>
> > **[Question 3]** l 279: "The participants are allowed to choose multiple best explanations only if the chosen ones are the same" Why? Can't two explanations be equally good (or bad)?
>
> This decision was suggested by a consultant at the data labeling company. We previously allowed multiple choices as best explanations, but the consultant reported that labelers found it unclear whether two explanations are equally good. As a result, it is difficult to determine if multiple explanations should all be considered “best.” We were told this option led to confusion and lower agreement among labelers. We updated the guidelines to allow only one best explanation, as we had already collected multiple choices when analyzing the percentage of good explanations. This will be explained in the paper.
>
> > **[Quality Weakness 5]** As for related work, the authors write: "Collecting ground-truth labels of human decision processes are quite expensive, if not impossible." But there exist a number of ML approaches that do exactly that - using post-hoc and self-explainable models. See for instance … just to mention a few. I would expect there to be also works on acquiring rules from human annotators in the context of logical and neuro-symbolic models, but I cannot provide pointers. I would have expected better coverage of works on debugging neural networks and self-explainable models.
>
> Thank the reviewer for pointing this out. We agree that this sentence is misleading and will revise it to “Collecting ground-truth labels of human decision processes for *every input instance is expensive and limits the method’s scalability*.” We hope to clarify here that **the goal of this paper is consistent with the papers suggested by the reviewer**: to assist humans in integrating feedback into the model. To achieve this goal, we proposed that most explanations should have good local coherency (human precision) prior to human intervention, so that humans are not required to correct the explanation for each instance, which is **expensive and not scalable**. Using our method, users only need to focus on the most problematic part, as demonstrated in Section 3.3’s user studies and model steering case study.
>
> >Other Comments
> > - l 121: "The deep model f is used to encode the sample x" Wasn't f the complete model? Also, at this point it is not clear what a "recursive generation layer" is. I'd suggest to move lines 121-123 to a later section.
> > - l 183: square brackets, missing comma
> > - l 191: organizes
> > - Eq 3. The equality symbols should be replaced by \approx, especially the second one (as p is replaced by \hat{p})
>
> We appreciate the insightful and constructive feedback. $f$ is the complete model, and we use all layers of the deep model f except for the softmax function to encode the sample $x$. We will revise the paper based on your suggestions.
>
> > l 125 : "Our framework for deep logic rule reasoning bears an interesting resemblance to the human mind..." All concept-based models and many neuro-symbolic models do. But it's only a vague analogy and I'm not sure what it adds. Moreover, notice that post-hoc models bear a striking resemblance to models of the mind in which decisions are taken subconsciously and then interpreted after the fact by an argumentative interpreter (see books by Michael Gazzaniga). There is no conclusive evidence of how our minds really work in this regard.
>
> Thank you for bringing up this interesting discussion. It is true that many models have a similar structure, i.e., an unexplainable deep model part and an explainable and conscious reasoning part. **Here, we are not trying to say that we are unique**. Instead, we wish to argue that having an unexplainable deep learning part in a self-explaining model is acceptable, since the human mind also has a subscious part that is not explainable. We will refine this part to improve clarity and avoid confusion.
>
> > is really the subconscious mind "often thought to be more powerful"?
>
> There is some evidence that the subconscious mind may be more powerful, such as “Charles Darwin depicted that subconscious mind is more powerful than conscious mind” [1] and “subconscious language acquisition has been shown to be more powerful than conscious learning” [2]. However, we agree that this statement may be controversial, and whether it is true largely depends on how we define “powerful.” Thus, we will remove this claim from the paper, since it is not the main focus of the paper.
>
> [1] Visual content search using subconscious retinal response. IEEE Region 10 Humanitarian Technology Conference 2017
>
> [2] Is CALL obsolete? Language acquisition and language learning revisited in a digital age. Tesl-Ej. 2014

---

> > ### Comment · Reviewer_YUk5 · 2022-08-07
> > **Reply to 6/6**
> >
> > > This decision was suggested by a consultant at the data labeling company. We previously allowed multiple choices as best explanations, but the consultant reported that labelers found it unclear whether two explanations are equally good.
> >
> > I find this a bit odd - I suppose that user confusion could still be there, but kind of hidden by the fact that users are required to pick one option.  Still, I do not have strong evidence against this choice, and I will refrain to take this aspect of the experiment into consideration in my review.
> >
> > > We agree that this sentence is misleading and will revise it to “Collecting ground-truth labels of human decision processes for every input instance is expensive and limits the method’s scalability.
> >
> > Fine by me.
> >
> > > Here, we are not trying to say that we are unique.
> >
> > Okay, I was under the impression that this was the intended meaning of that paragraph.  I am still a bit puzzled about the contribution of this paragraph to the overall message of the paper and I would prefer if it were removed.  But I realize that this is partly a matter of preferences.  Not a huge deal either way.

---

> > > ### Author Response · Authors · 2022-08-08
> > > **Additional Response to Reviewer YUk5 about Reply to (6/6)**
> > >
> > > Thanks a lot for your thoughtful consideration. We will remove the paragraph about the resemblance to the human mind from the paper to avoid confusion, and more carefully handle the participants’ confusion in our future user studies.

---

### Official Review · Reviewer_zwRN · 2022-07-11

**Rating:** 6
**Confidence:** 4
**Soundness:** 3 good
**Presentation:** 3 good
**Contribution:** 2 fair

**Summary:**

This paper proposes a self-explaining model generating logic rules as explanations. It aims to learn explanations that (1) make accurate prediction, (2) are coherent with human's prior belief. The authors provides several experimental evidences supporting their claims using three benchmark datasets (two text datasets and a tabular dataset). The major contribution is on the quality of explanation where their method achieved a good performance compared to the others. But the evaluation of explanation quality is limited to human precision. It seems like the authors perform several other experiments (C.4.4, C.4.3) that are potentially related with explanation quality, but didn't discuss their experimental findings or relate with the explanation quality. I expect the paper can be improved during the rebuttal period, and am willing to adjust my score if it addresses several concerns listed below.

**Questions:**

### Majors
(1)  Figure 1c: What are the assumptions you made for the generative process? For example, see the following questions:
- Do you assume that the antecedents (explanations) and corresponding 1st order logic rules are sufficient for making accurate prediction?
- Do you assume that given $\alpha$, $\mathbf{x}$ and $b$ are independent? What is rationale for this?

(2) Provide AUCs for Table 2.

(3) Provide details about the "Base" model. It is usually expected model performance degrades when an explanation mechanism is added to the model. It is important to compare with reasonably well designed baseline models to precisely evaluate classification performance of explainable models. (Btw, it is sometimes good enough to have comparable prediction performance with prediction-only baseline models, if it can provide substantially good quality of explanations in different aspects.)

(4) The following questions are related with the quality of explanations.
- Are there any factors (e.g., rule length, number of atoms) impacting explanation quality? Do you expect trade-off between prediction performance and quality of explanation?
- How unique the explanations are for an instance (sample)? (i.e., do the models trained on two different random seeds generate the same or similar logic rules for an instance?) Is there any factor controlling the uniqueness? -- I used the term "uniqueness" instead of "robustness" because the authors already used the term "robustness" for another purpose in the paper.

### Minors
- Provide standard error of the five runs for the classification performance in Table 2. (Figure 8 is unnecessary).
- Fix typo in the equation (1)
- No content in C.4.2

**Limitations:**

First, it should discuss how likely the assumptions made for Figure 1c are correct in practice, and what we need to expect when they're not satisfied (see Q1). Second, it should discuss that the explanation quality can depend on several factors (see Q4), and careful designed experiments are required to get good enough explanation.


**Strengths And Weaknesses:**

### Quality
Strength
- If we assume Figure 1c is well designed process, the model component design itself has a good rationale. The model is composed of three parts (human prior, consequent estimation, antecedent generation) with corresponding loss terms. Each part takes a role to achieve the coherence, prediction, and generation.
- The authors' main claims (prediction accuracy, and human precision) are supported by various experiments including human evaluation.

Weakness
- The generative process (Figure 1c) should be provided with a good rationale, or in-depth discussion why it makes sense. [See Q1]
- Explanation quality should be evaluated in different aspects. (It is important especially when the prediction performance are not substantially good enough.) [See Q4]

### Clarity
Weakness
- Lack of details about important experimental settings: "Base" model [See Q3]


### Significance
Strength
- The authors' main claims (prediction accuracy, and human precision) are supported by various experiments including human evaluation. Especially, it achieves a good performance on the human precision evaluation.
- In addition to the main experimental results, the paper provides several interesting experimental observations about relationships between model interpretability and robustness, and potentiality for debugging and refinement using the model explanations. Although those are not based on in-depth or wide-breadth experiments (probably because this is not directly related with the authors' main claims), those observations can fuel interesting future works in this direction.

Weakness
- It seems like there are no significant improvement in the classification performance (Table 2).

---

> ### Author Response · Authors · 2022-08-02
> **Response to Reviewer zwRN [1/4]**
>
> > Assumptions made for the generative process and their rationale
> > - **[Quality Weakness 1]** The generative process (Figure 1c) should be provided with a good rationale, or in-depth discussion why it makes sense. [See Q1]
> > - **[Major Question 1]**  Figure 1c: What are the assumptions you made for the generative process? For example, see the following questions:
> > - - Do you assume that the antecedents (explanations) and corresponding 1st order logic rules are sufficient for making accurate prediction?
> > - - Do you assume that given α, x and b are independent? What is rationale for this?
> > - **[Limitation 1]** First, it should discuss how likely the assumptions made for Figure 1c are correct in practice, and what we need to expect when they're not satisfied (see Q1).
>
> Thank you for raising this fundamental issue. We agree that the underlying assumptions of the generative process should be clarified early in the paper. Please find our assumptions and rationales as follows. We will refine the introduction to Eq. (1) to better explain them accordingly.
>
> **(Assumption A)** For $p(y|x,b)=\sum_{\alpha}{p(y|\alpha)p(\alpha|x,b)}$, we assume that $p(y|\alpha)=p(y|\alpha,x,b)$.
>
> This can be decomposed into two assumptions: $p(y|\alpha)=p(y|\alpha,x)$ (A1) and $p(y|\alpha,x)=p(y|\alpha,x,b)$ (A2).
>
> Assumption A1 $p(y|\alpha)=p(y|\alpha,x)$ indicates that **explanation $\alpha$ contains all of the information in input $x$ that is used to predict $y$**.
> This formulation compels the model to pass information from $x$ to $y$ only via explanations, as opposed to other unexplainable parts. This assumption may limit the prediction performance, but it is **essential for $\alpha$ to be a trustable explanation** for predicting $y$.
> Otherwise, there may be a direct connection between $y$ and $x$ that is unrelated to the explanation $\alpha$. Thus, $\alpha$ may only explain a small portion of the model behavior (e.g., only explain 1% of the change in $y$), and differ substantially from the ground-truth explanation of the overall model behavior.
>
> Assumption A2 $p(y|\alpha,x)=p(y|\alpha,x,b)$ means that  explanation $\alpha$ and input $x$ contain all of the information in $b$ (human prior preference for explanations) that can be used to predict $y$. It is intuitive that this assumption holds, as human preference for explanations is unrelated to the current class label.
>
> **(Assumption B)** For $\sum_{\alpha}{p(y|\alpha)p(\alpha|x,b)}\propto\sum_{\alpha}{p(b|\alpha)p(y|\alpha)p(\alpha|x)}$, we assume that $p(x, b)=p(b)p(x)$ and $p(x, b|\alpha)=p(b|\alpha)p(x|\alpha)$ (see supplement B, probability decomposition, for a proof).
>
> It means that **$x$ and $b$ are independent** no matter whether $\alpha$ is given. In other words, **seeing input sample ($x$) does not change the belief in our prior preference for explanations ($b$)**, no matter whether the explanations $\alpha$ are given or not, i.e., $p(b)=p(b|x)$ and $p(b|)=p(b|x,\alpha)$. The rationale for the assumption is that our prior preferences for explanations are usually fixed and unrelated with the input $x$. Even if this assumption is not satisfied, it will not have a significant effect on our framework. Only the human prior module must be integrated into the antecedent generation module, which changes from $p(\alpha|x)$ to  $p(\alpha|x,b)$.

---

> > ### Comment · Reviewer_zwRN · 2022-08-07
> > **Thank you for the responses**
> >
> > Thank you for the responses and additional experiments to address my concerns. I updated my score to 6 (Weak Accept).
> >
> > I see most of the experimental results are intuitively understandable, except the one about the rule length. Usually, a longer rule provides a more complicated explanation, so people may think it is less explainable. However, the experimental results indicate longer rules provide a better explanation. The authors should be careful when making the argument like 'longer rules tend to improve human precision. This indicates longer rules contain more useful information for decision making, resulting in greater precision.'

---

> > > ### Author Response · Authors · 2022-08-08
> > > **Additional Response to Reviewer zwRN**
> > >
> > > >Concern about rule length
> > > > - I see most of the experimental results are intuitively understandable, except the one about the rule length. Usually, a longer rule provides a more complicated explanation, so people may think it is less explainable. However, the experimental results indicate longer rules provide a better explanation. The authors should be careful when making the argument like 'longer rules tend to improve human precision. This indicates longer rules contain more useful information for decision making, resulting in greater precision.'
> > >
> > > Thank you for carefully checking our response and the thoughtful suggestion. Here we explain more about the user study result:
> > >
> > > * The length of rule is actually the *maximum* length of the rule: our method can automatically generate shorter rules than the default length. This is achieved by allowing the model to select the NULL atom which is always true. Sorry for the misleading description.
> > >
> > > * We evaluate *human precision* here instead of readability or cognitive load. Thus we study which method predicts by using features most similar to that used by humans. To this end, we define a good explanation as one that naturally leads to the prediction, although it may contain noisy features. The best explanation is the one that contains the most important features and least noisy features for decision making. A longer rule may have a larger percentage of good because it has more chances to find a good atom (feature), and the fact that longer rules have a larger percentage of best indicates that our model adds additional important features (and less noisy features) when rule length increases.
> > >
> > > We will carefully revise our text to avoid any misconception and confusion.

---

> ### Author Response · Authors · 2022-08-02
> **Response to Reviewer zwRN [2/4]**
>
> > Better evaluation of explanation quality.
> > - **[(Quality Weakness 2]** Explanation quality should be evaluated in different aspects. (It is important especially when the prediction performance are not substantially good enough.) [See Q4]
> > - **[Major Question 4]** The following questions are related with the quality of explanations.
> > - - Are there any factors (e.g., rule length, number of atoms) impacting explanation quality? Do you expect trade-off between prediction performance and quality of explanation?
> > - - How unique the explanations are for an instance (sample)? (i.e., do the models trained on two different random seeds generate the same or similar logic rules for an instance?) Is there any factor controlling the uniqueness? -- I used the term "uniqueness" instead of "robustness" because the authors already used the term "robustness" for another purpose in the paper.
> > - **[Limitation]** Second, it should discuss that the explanation quality can depend on several factors (see Q4), and careful designed experiments are required to get good enough explanation.
>
> We thank the reviewer for raising this important issue and guiding us to address it. Per your suggestion, we hired 6 participants and asked them to label the explanations generated with different rule length and number of atoms. We will add this result in the paper.
>
> Tables 1 and 2 show how human precision of explanations change with **rule length**. We can see that rules of all lengths, even short rules with only one atom, have a certain level of explainability (e.g., the average percentage of good for Length 1 rule is 79.7%). Meanwhile, longer rules tend to improve human precision. This indicates longer rules contain more useful information for decision making, resulting in greater precision. Here, we marked one star (*) if the p-value is lower than 0.1, and two stars (**) if it is less than 0.05.
>
> **Table 1 Percentage of Good Explanations with Varying Rule Length (Yelp). P1 to P6 represent 6 participants.**
> |  | Length 1 | Length 2 | Length 3 | Length 4 |
> |---|---|---|---|---|
> | P1 | 76.0 | 92.0 | 96.0 | 100.0 |
> | P2 | 74.0 | 90.0 | 94.0 | 100.0 |
> | P3 | 76.0 | 86.0 | 88.0 | 90.0 |
> | P4 | 76.0 | 94.0 | 92.0 | 98.0 |
> | P5 | 96.0 | 100.0 | 100.0 | 100.0 |
> | P6 | 80.0 | 84.0 | 84.0 | 86.0 |
> | Average | 79.7 | 91.0 | 92.3 | **95.6** |
> | Agreement | 87.6 | 91.3 | 89.7 | 91.9 |
> | P-value (compared with length 4) | 1.31 E-11 ** | 8.93 E-4 ** | 3.73 E-3 ** | N/A  |
>
>
> **Table 2 Percentage of Best Explanations with Varying Rule Length (Yelp). P1 to P6 represent 6 participants.**
> |  | Length 1 | Length 2 | Length 3 | Length 4 |
> |---|---|---|---|---|
> | P1 | 8.0 | 24.0 | 20.0 | 56.0 |
> | P2 | 4.0 | 26.0 | 28.0 | 50.0 |
> | P3 | 2.0 | 10.0 | 18.0 | 68.0 |
> | P4 | 6.0 | 12.0 | 28.0 | 62.0 |
> | P5 | 10.0 | 14.0 | 14.0 | 72.0 |
> | P6 | 8.0 | 10.0 | 26.0 | 58.0 |
> | Average | 6.3 | 16.0 | 22.3 | **61.0** |
> | Agreement | 94.3 | 81.6 | 81.7 | 66.5 |
> | P-value (compared with length 4) | 2.38 E-43 ** | 6.98 E-23 ** | 1.58 E-16 ** | N/A |
>
> *(not finished, see the next message: response [3/4])*

---

> ### Author Response · Authors · 2022-08-02
> **Response to Reviewer zwRN [3/4]**
>
> Tables 3 and 4 show how human precision of explanations change with **the number of candidate atoms**. In particular, 1000 means that we use the top 1000 frequent words as the candidate atoms. In Tables 3 and 4, we could find that explanation quality increases with increasing number of atoms, when the number of atoms is no larger than 1000. After that (e.g., when the number of atoms is 5000 and 10000), there is no statistically significant difference in explainability. This demonstrates that our model requires a minimum of approximately 1000 atoms to provide a good explanation. This finding aligns with the observations in [1], i.e., analyzing and explaining common content such as restaurant reviews and news articles does not require a large vocabulary.
>
> [1] How many words do you need to speak a language? BBC, https://www.bbc.com/news/world-44569277
>
> **Table 3 Percentage of Good Explanations with Varying Number of Atoms (Yelp). P1 to P6 represent 6 participants.**
> | # Atoms | 10 | 100 | 1000 | 5000 | 10000 |
> |---|---|---|---|---|---|
> | P1 | 10.0 | 34.0 | 84.0 | 100.0 | 96.0 |
> | P2 | 12.0 | 40.0 | 82.0 | 98.0 | 96.0 |
> | P3 | 14.0 | 46.0 | 86.0 | 98.0 | 94.0 |
> | P4 | 8.0 | 34.0 | 78.0 | 90.0 | 88.0 |
> | P5 | 26.0 | 64.0 | 100.0 | 100.0 | 100.0 |
> | P6 | 24.0 | 58.0 | 80.0 | 86.0 | 90.0 |
> | Average | 15.7 | 46.0 | 85.0 | **95.3** | 94.0 |
> | Agreement | 94.5 | 94.5 | 94.1 | 91.7 | 91.7 |
> | P-value (compared with 5000) | 5.86 E-2 * | 5.86 E-2 * | 1.58 E-1 | N/A  | 2.06 E-1 |
>
> **Table 4 Percentage of Best Explanations with Varying Number of Atoms (Yelp)**
> | # Atoms | 10 | 100 | 1000 | 5000 | 10000 |
> |---|---|---|---|---|---|
> | P1 | 0.0 | 6.0 | 30.0 | 54.0 | 46.0 |
> | P2 | 2.0 | 6.0 | 38.0 | 56.0 | 48.0 |
> | P3 | 4.0 | 4.0 | 32.0 | 56.0 | 52.0 |
> | P4 | 0.0 | 6.0 | 30.0 | 52.0 | 46.0 |
> | P5 | 0.0 | 4.0 | 44.0 | 48.0 | 48.0 |
> | P6 | 0.0 | 2.0 | 34.0 | 50.0 | 46.0 |
> | Average | 1.0 | 4.7 | 34.7 | **52.7** | 47.7 |
> | Agreement | 77.1 | 76.9 | 73.5 | 71.2 | 74.3 |
> | P-value (compared with 5000) | 6.25 E-5 ** | 8.16 E-5 ** | 2.45 E-3 ** | N/A | 2.20 E-1 |
>
> Tables 5 and 6 illustrate **how explanation quality changes with the prediction performance**. There is no evidence for a trade-off between explainability and prediction performance. In contrast, **models with good explainability are also models with good prediction performance** (i.e., models with rule length 2 to 4 or number of atoms 1000 to 10000). This is consistent with our framework $p(y|x,b)=\sum_{\alpha}{p(y|\alpha)p(\alpha|x,b)}$, which passes information from input $x$ to prediction $y$ only via explanations, as opposed to other unexplainable parts. Thus, the expressivity of the explanations and the capacity of the model are tightly related. If the hyperparameter settings significantly constrain the expressivity of the explanations (e.g., number of atoms 10), both explanation quality and predictive performance will decrease significantly.
>
> **Table 5 Comparison between Prediction Performance and Explanation Quality with Varying Rule Length**
> |  | Length 1 | Length 2 | Length 3 | Length 4 |
> |---|---|---|---|---|
> | Ratio of Good Explanations | 79.7 | 91.0 | 92.3 | **95.6** |
> | Ratio of Best Explanations | 6.3 | 16.0 | 22.3 | **61.0** |
> | Prediction Performance (F1) | 96.13 | **96.47** | 96.32 | 96.26 |
> | Prediction Performance (ROC AUC) | 98.05 | 98.48 | 98.54 | **99.41** |
>
> **Table 6 Comparison between Prediction Performance and Explanation Quality with Varying Number of Atoms**
> | # Atoms | 10 | 100 | 1000 | 5000 | 10000 |
> |---|---|---|---|---|---|
> | Ratio of Good Explanations | 15.7 | 46.0 | 85.0 | **95.3** | 94.0 |
> | Ratio of Best Explanations | 1.0 | 4.7 | 34.7 | **52.7** | 47.7 |
> | Prediction Performance  (F1) | 33.3 | 95.15 | 96.24 | **96.26** | 96.14 |
> | Prediction Performance (ROC AUC) | 87.57 | 97.60 | 98.48 | **99.41** | 98.31 |
>
> We also conduct experiments to confirm that our model usually generates **unique explanations** for the same instances.  Comparing the model explanations trained with 5 seeds reveals that, on average, **90.04%** of atoms were shared by explanations from different seeds, and **71.27%** were identical on Yelp. This comparison suggests that our model generates a unique explanation for the same instance, even **in the absence of a direct controlling factor**. The reason why we can generate unique explanations is that we optimize the explanation generator with two globally consistent rewards (Eq. (6)): 1) human’s prior belief about which explanation types are good and 2) the explanation (rule) confidence that is measured by the global prediction accuracy over the entire training corpus given the rule. Since the second reward is a real number instead of a discrete value and has a globally consistent meaning, the optimal explanation is usually unique and stable, leading to similar results when trained with different random seeds.

---

> ### Author Response · Authors · 2022-08-02
> **Response to Reviewer zwRN [4/4]**
>
> > Details about the “Base” model
> > - **[Clarity Weakness]** Lack of details about important experimental settings: "Base" model [See Q3]
> > - **[Major Question 3]** Provide details about the "Base" model. It is usually expected model performance degrades when an explanation mechanism is added to the model. It is important to compare with reasonably well designed baseline models to precisely evaluate classification performance of explainable models. (Btw, it is sometimes good enough to have comparable prediction performance with prediction-only baseline models, if it can provide substantially good quality of explanations in different aspects.)
>
> Our base model refers to commonly-used deep learning models without explainability. For textual datasets, we used Transformer-based pretrained models, **BERT** (bert-base-uncased) and **RoBERTa** (roberta-base), as the backbone and a single fully connected layer that predicts probability for each class from the instance embedding. For tabular datasets, a 3-layer **DNN (fully connected layers)** is employed with ReLU activation function between layers (FC-ReLU-FC-ReLU-FC). Another fully connected layer is leveraged for classification after the DNN layers. Our self-explainable models are built by using the corresponding base model as the backbone. Our experiment results show that we have comparable prediction performance with prediction-only baseline models, as you suggested. We will make it clearer in the paper.
>
>
>
> > **[Significance Weakness]** It seems like there are no significant improvement in the classification performance (Table 2).
>
> It is true that our classification performance has not improved significantly. Compared to existing unexplainable and self-explaining models, we consider our contribution to be achieving substantially better explainability and robustness while maintaining or sometimes improving classification performance. To clarify this, we will revise the abstract and introduction.
>
>
> > **[Major Question 2]** Provide AUCs for Table 2.
>
> Table 7 shows the result of AUC. Here, we evaluate the PR AUC instead of the ROC AUC because the latter is less suitable for imbalance data [1][2]. Note that the Clickbait dataset is highly imbalanced (news:clickbait = 4:1). The numbers have been changed from the initial response. We provide the average and standard errors of 5 runs instead of a single run.
>
> [1] F1 Score vs ROC AUC vs Accuracy vs PR AUC: Which Evaluation Metric Should You Choose? - neptune.ai
>
> [2] The precision-recall plot is more informative than the ROC plot when evaluating binary classifiers on imbalanced datasets. PloS one. 2015 Mar 4;10(3):e0118432.
>
> **Table 7 Classification Performance (AUC). The numbers in brackets are standard errors.**
> |  | Yelp |  | Clickbait |  | Adult |
> |---|:---:|:---:|:---:|:---:|:---:|
> |  | BERT | RoBERTa | BERT | RoBERTa | DNN |
> | Base | 97.39 (0.0659) | 97.90 (0.0577) | 62.27 (1.0400) | 63.72 (0.8722) | 68.62 (0.2317) |
> | SENN | 96.00 (0.1087) | 96.97 (0.0841) | 55.64 (1.0118) | 57.93 (0.7779) | 67.39 (0.0854) |
> | RCN | 97.31 (0.0274) | 98.03 (0.0086) | 59.91 (0.2024) | 59.37 (0.2259) | 70.06 (0.0411) |
> | Ours | 97.28 (0.0335) | 97.78 (0.0833) | 60.31 (0.8498) | 64.14 (0.5906) | 70.36 (0.0892) |
>
> > **[Minor Question 1]** Provide standard error of the five runs for the classification performance in Table 2. (Figure 8 is unnecessary).
>
> Thanks for the suggestions, we will modify the paper accordingly. Table 8 contains the prediction performance measured on F1 score and its standard error.
>
>
> **Table 8 Classification performance (F1). The numbers in brackets are standard errors.**
> |  | Yelp |  | Clickbait |  | Adult |
> |---|---|---|---|---|---|
> |  | BERT | RoBERTa | BERT | RoBERTa | DNN |
> | Base | 96.20 (0.0541) | 97.16 (0.0672) | 72.84 (0.9302) | 74.25 (0.7763) | 76.15 (0.2522) |
> | SENN | 95.12 (0.1995) | 96.07 (0.1180) | 69.09 (0.9550) | 70.99 (0.5076) | 71.69 (0.7681) |
> | RCN | 96.38 (0.0089) | 97.36 (0.0049) | 68.80 (0.1359) | 68.64 (0.1467) | 77.35 (0.0309) |
> | Ours | 96.26 (0.0445) | 97.13 (0.0642) | 71.12 (0.5479) | 74.20 (0.5009) | 77.37 (0.2374) |
>
>
>
> > **[Minor Questions 2, 3]**
> > - Fix typo in the equation (1)
> > - No content in C.4.2
>
> Thank the reviewer for these suggestions. We will carefully modify the paper according to the suggestions.

---

### Author Response · Authors · 2022-08-09
**Special Thanks to All Reviewers About Their Insightful Comments**

We sincerely thank all reviewers for your deep understanding of our method, the expertise you possess, the constructive, considerate and insightful suggestions, and the patience you demonstrated when reading our lengthy responses. The depth and quality of the discussions have greatly exceeded our expectations, which we have seldom experienced before and has significantly improved the readability and soundness of our paper. We will strictly follow our promise to revise the final version of the paper and carefully incorporate the findings we obtained during the discussion. Thanks a lot for all of your help and wish you all the best in your research!

---

### Meta-Review · Area_Chair_izjw · 2022-08-25

**Recommendation:** Accept
**Confidence:** Less certain

**Metareview:**

The paper deals with the important topic of devising accurate predictive models that are able to distill explainations that can be easily understood by humans. In particular, authors propose a deep neural network that predicts logical rules over which a human-specified prior can be imposed.

The reviewers agreed that the scope of the contribution is relevant and the contribution is timely. At the same time, they highlighted some shortcomings concerning the experimental setting (e.g. some metrics or baselines missing), the motivation and effect of certain assumptions over the explainations (e.g., are rules consistent).
During the rebuttal, the authors managed to address the aforementioned concerns in a satisfactory way which saw some scores improve (kudos!)

The paper is accepted conditioned on the inclusion of all the experimental material and discussion that emerged during the rebuttal.

**Award:**

No

---

### Decision · Program_Chairs · 2022-09-14

Accept